# Multi-Modal Graph Neural Networks for Colposcopy Data Classification and Visualization

**DOI:** 10.3390/cancers17091521

**Published:** 2025-04-30

**Authors:** Priyadarshini Chatterjee, Shadab Siddiqui, Razia Sulthana Abdul Kareem, Srikanth R. Rao

**Affiliations:** 1Department of Computer Science and Engineering, Koneru Lakshmaiah Education Foundation, Hyderabad 500075, Telangana, India; cseshadabsiddiqui@gmail.com; 2Old Royal Naval College, University of Greenwich, Park Row, London SE10 9LS, UK; razia.sulthana@greenwich.ac.uk; 3MNJ Institute of Oncology and Regional Cancer Center, Hyderabad 500004, Telangana, India; srikanthsapthagiri@yahoo.com

**Keywords:** cervical lesion classification, graph neural networks (GNNs), hyperparameter optimization, multi-modal data integration

## Abstract

Cervical lesion classification plays a crucial role in the early detection of cervical cancer, but conventional deep learning methods often depend on single-modal data or extensive manual annotations. This study proposes a graph-based multi-modal learning framework that integrates colposcopy images, segmentation masks, and graph representations to improve classification accuracy. A fully connected Graph Neural Network (GNN) architecture was developed and evaluated through five-fold cross-validation and fine-tuning. The model achieved a macro-average F1-score of 94.56% and a validation accuracy of 98.98% after fine-tuning. Visual explanations confirmed the model’s ability to focus on relevant lesion regions. The framework, validated in collaboration with the MNJ Institute of Oncology, shows strong potential for clinical deployment.

## 1. Introduction

Cervical cancer remains a global health challenge, with over 604,000 new cases and 342,000 deaths reported in 2020 [1]. Despite its slow progression, early screening can reduce incidence by over 60% [2,3]. Standard cytology-based techniques, such as Pap smears and the ThinPrep Cytology Test (TCT), follow criteria like The Bethesda System (TBS) [4,5] but face limitations in diagnostic consistency due to dependence on expert interpretation and labor-intensive processes.

Deep learning has shown promise in automating cervical lesion classification, typically using a segmentation–classification pipeline [6,7]. However, segmentation quality directly impacts classification accuracy [8]. To address this, we propose a novel multi-modal graph-based framework combining colposcopy images, segmentation masks, and Graph Neural Networks (GNNs) to improve both feature representation and interpretability.

Previous studies relied heavily on labeled cytology datasets and single-modal inputs [9,10,11,12], limiting generalizability. Our approach incorporates spatial and structural lesion characteristics using graph representations, reducing annotation dependency [13,14,15]. Unlike semi-supervised models, which often overlook spatial dependencies [16], our method captures lesion topology and complex morphology critical for grading.

Robustness is ensured via K-fold cross-validation and grid search for hyperparameter tuning [17,18,19], addressing overfitting and variability across imaging domains [20]. Colposcopy presents unique challenges, such as lighting variation and tissue morphology artifacts, which conventional CNNs struggle to resolve. We mitigate this through DeepLabv3-based segmentation and GNN-based spatial reasoning, closely emulating pathologist assessment.

In this study, we aim to develop a graph-integrated multi-modal model to improve cervical lesion detection by enhancing interpretability, accuracy, and clinical applicability. Extensive experiments on labeled colposcopy datasets, validated through rigorous cross-validation and hyperparameter tuning, demonstrate that our graph-integrated multi-modal model outperforms traditional classification approaches [21]. By leveraging graph-based reasoning and structural learning, our model enhances interpretability, accuracy, and clinical applicability in cervical lesion detection.

(1)**Multi-Modal Graph Integration:** We propose a GNN-based framework integrating colposcopy images, DeepLabv3-generated segmentation masks, and structural graph representations for enhanced cervical lesion classification.(2)**Feature-Driven Segmentation:** Lesion features such as area, perimeter, eccentricity, and GLCM-based textures are extracted to support discriminative graph construction and clinically interpretable outputs.(3)**Graph-Based Reasoning:** Graphs are constructed using Euclidean similarity-weighted edges, capturing lesion topology and improving decision consistency across subtle morphological variations.(4)**Robust Validation and Optimization:** The framework employs five-fold cross-validation and grid search to ensure high reliability and mitigate overfitting.(5)**Cross-Domain Adaptability:** Evaluations on brain tumor MRI, eye_disease, and Malhari colposcopy datasets demonstrate the model’s generalizability to other medical imaging tasks.(6)**Clinically Interpretable Outputs:** The system generates segmentation overlays, graph structures, and LIME visualizations with metrics and class predictions, aiding clinician trust and decision support.

## 2. Literature Review

1.Graph-Based Medical Image Classification Methods

Graph-based deep learning has significantly advanced medical image analysis by capturing complex structural relationships, particularly in non-Euclidean data representations. Early methods such as GCNN [22] and GCN [23] introduced convolutional operators tailored for graph structures, yet these methods suffer from over-smoothing, leading to feature dilution in deeper layers.

To address this, GAT [24] incorporates attention mechanisms for adaptive feature aggregation, demonstrating improved classification accuracy but at the cost of increased computational complexity. Hybrid approaches, such as GCNN-EC [25] and CCF-GNN [26], integrate graph connectivity enhancements, enabling better feature propagation and hierarchical representation learning.

Transformer-based models like AGT [27] and GRM [28] further extend GNN capabilities by leveraging self-attention for long-range dependency modeling; however, their reliance on large-scale labeled data and computational overhead limit their practical applicability.

Hypergraph models [29] aim to capture higher-order correlations in neuron segmentation but require significant memory resources, making them inefficient for large-scale medical datasets. Few-shot graph models [30] and multi-modal frameworks such as MRCG [31] attempt to address data scarcity through self-supervised learning, but dynamic graph construction remains an open challenge. Existing multi-modal approaches [32] incorporate segmentation masks, histopathological features, and topological representations, yet most fail to generalize across diverse datasets.

2.Multi-Modal Data Integration

While graph-based models have demonstrated significant improvements in medical image classification by capturing spatial relationships and non-Euclidean data structures, their performance can be further enhanced by integrating multi-modal information. Single-modal approaches, such as nonrigid registration [33], hypergraph neural-like P systems [34], and GCNN-EC [35], have proven effective for specific imaging tasks but lack the ability to leverage complementary data sources for improved classification.

Recent advances in multi-modal learning have aimed to address these limitations by incorporating additional modalities such as MRI retrieval [36], knee cartilage defect assessment [37], and histopathological analysis [38]. However, these approaches are often constrained to segmentation and retrieval tasks, limiting their effectiveness in predictive classification.

Furthermore, the challenge of insufficient paired multi-modal data remains critical, motivating the use of generative models such as Conditional CycleGAN for data augmentation. By leveraging multi-modal fusion techniques and generative strategies, recent works aimed to enhance diagnostic precision and address missing data issues, thereby improving classification performance beyond single-modal baselines [39].

Other studies, such as those on bi-graph reasoning for feature fusion [40] and few-shot PPE detection [41], attempted to enhance classification performance through feature-level integration, yet they lack generalization across diverse datasets.

Furthermore, multi-label proxy metric learning [42] provides a means of optimizing classification boundaries, but many of these methods fail to incorporate robust validation strategies such as K-fold cross-validation and hyperparameter tuning [33,35,38].

3.Validation and Hyperparameter Optimization

Building upon the advancements in graph-based learning and multi-modal data integration, ensuring model generalizability and robustness remains a critical challenge in medical image classification. While single-modal methods such as the Enhanced HBO Algorithm [43] have demonstrated success in hyperparameter optimization, they lack multi-modal validation, making them unsuitable for complex medical datasets.

GAN-based frameworks [44] and Kernel Method Encoders [45] enhance feature extraction, yet the absence of K-fold cross-validation limits their applicability across diverse clinical datasets. More recently, deep learning architectures such as TS-CNN [46] and PubMedCLIP [47] sought to improve model interpretability, but they still fail to incorporate systematic hyperparameter tuning.

The Semi-Supervised Quadratic Neurons model [48] attempts to improve robustness but lacks the advanced cross-validation techniques required for fine-tuned optimization. Similarly, task-specific models such as MTANet [49] and Simi-Attention [50] excel in segmentation but overlook iterative tuning and validation strategies.

Despite recent improvements in uncertainty quantification, models like Hercules [51] still do not leverage systematic hyperparameter optimization, limiting their adaptability in real-world applications.

Federated learning techniques such as Adaptive Aggregation [52] address dataset variability but lack rigorous validation across multiple clinical environments. Furthermore, models like DASQE [52,53] and TPAS [51,54] focus on managing data uncertainty yet fail to integrate robust K-fold cross-validation.

4.Classification Models without Segmentation

Following the advancements in graph-based classification, multi-modal data integration, and hyperparameter optimization, many existing classification models still lack segmentation capabilities, which are crucial for enhanced spatial and contextual learning. While models like that in [55] propose optimization strategies for classification, they do not incorporate segmentation for localized feature extraction.

Similarly, self-interpretable CNN architectures [56] improve interpretability but fail to leverage segmentation for spatial insights. Quadratic neuron-based approaches [57] enhance classification accuracy but overlook spatial data integration, limiting their robustness.

GAN-based augmentation techniques [58] improve image quality but do not integrate segmentation as a feature enhancement step.

Moreover, multi-modal learning models such as PubMedCLIP [59] focus on feature fusion yet fail to incorporate spatial segmentation. Kernel-based classification strategies [60] also lack segmentation capabilities, reducing their effectiveness in medical image analysis.

Additionally, dual feature extraction models like that in [61] prioritize feature extraction but remain constrained to classification tasks without segmentation integration.

## 3. Materials and Methods

In this section, we first provide an overview of our proposed framework. Following that, we introduce the process of segmentation mask generation and feature extraction. Finally, the graph construction and Graph-Neural-Network-based classification approach, along with the validation and optimization strategies, are detailed.

### 3.1. Methods

#### 3.1.1. Framework Overview

Our framework, shown in Figure 1, is designed for supervised learning, employing a Graph Neural Network (GNN) to classify medical images using graph representations derived from segmentation masks and associated features. The training pipeline includes the creation of graph-to-mask-to-image mappings, feature extraction from segmentation masks, graph construction, and GNN-based classification [22,23] with K-fold cross-validation and hyperparameter optimization. Below, we elaborate on the key components.

1.Overall Architecture: The framework takes segmentation masks, extracts region-specific features (e.g., area, perimeter, texture), and constructs graphs where nodes represent regions and edges capture spatial relationships. These graphs, along with their associated labels, are fed into a GNN for classification. Our pipeline ensures robust generalization by performing K-fold cross-validation and leveraging grid search for hyperparameter tuning.1.A batch of graphs is loaded along with their labels.2.The GNN processes node features and edge weights to generate predictions.3.A cross-entropy loss function is computed between the predictions and true labels.4.Gradients are computed and used to update model parameters via backpropagation.

Unlike semi-supervised approaches, our method does not rely on unlabeled data or pseudo-labeling. Instead, it achieves robust predictions through explicit graph-based feature representation and optimized supervised learning.

2.Mathematical Formulation

##### Loss Function

The model uses cross-entropy loss [62] to quantify the difference between predicted logits and the ground truth labels. Given a batch of *N* graphs, the loss is computed as in Equation (Equation 1):(1)$$L=-\frac{1}{N}\sum_{i=1}^{N}\sum_{c=1}^{C}y_{i,c}\cdot \log{\hat{P}}_{i,c}$$
where *C* is the number of classes, $y_{i,c}$ is the one-hot encoded ground truth for the *i*-th graph and *c*-th class, and ${\hat{P}}_{i,c}$ is the predicted probability for the *c*-th class. This loss function is particularly beneficial for our model as it is well suited for multi-class classification tasks. It encourages the model to output high-confidence predictions for the correct class while penalizing incorrect predictions more heavily. By operating on the logarithm of predicted probabilities, cross-entropy loss provides a smooth gradient, which helps to optimize the weights effectively during training. Additionally, the probabilistic nature of this loss aligns well with the softmax output layer of the model, ensuring that the predicted logits are normalized into probabilities, which makes the training process numerically stable and interpretable.

##### Graph Construction

Each segmentation mask is converted into a graph:–Nodes represent segmented regions, with features such as area, perimeter, eccentricity, etc., denoted as $f_{i}\in R^{d}$ for the *i*-th node.–Edges capture spatial or feature-based relationships between regions. The edge weight between nodes *i* and *j* is computed using Equation (Equation 2):(2)$$w_{ij}=\frac{1}{\mathrm{dist}(f_{i},f_{j})+\epsilon }$$

Here, dist is the Euclidean distance, and ϵ prevents division by zero.

##### Graph Neural Network

The GNN processes graph-structured data with two GCNConv layers followed by a fully connected layer for classification, represented in Equations (Equation 3)–(Equation 6):(3)$$h_{i}^{\left(1\right)}=\mathrm{ReLU}\left(\sum_{j\in N(i)}\frac{1}{\sqrt{d_{i}d_{j}}}W_{1}h_{j}^{\left(0\right)}\right)$$(4)$$h_{i}^{\left(2\right)}=\mathrm{ReLU}\left(\sum_{j\in N(i)}\frac{1}{\sqrt{d_{i}d_{j}}}W_{2}h_{j}^{\left(1\right)}\right)$$(5)$$h_{\mathrm{graph}}=\mathrm{GlobalMeanPool}\left(\{h_{i}^{\left(2\right)}\}\right)$$(6)$$y=\mathrm{Softmax}(W_{\mathrm{out}}h_{\mathrm{graph}})$$
where $h_{i}^{\left(0\right)}$ are the initial node features, $h_{i}^{\left(1\right)},h_{i}^{\left(2\right)}$ are the node embeddings after the first and second layers, $h_{\mathrm{graph}}$ is the graph-level representation after pooling, and y is the predicted logits.

##### K-Fold Cross-Validation

To ensure robustness, the dataset is split into *K*-folds [43]. For each fold, the following apply:–K−1 folds are used for training, and the remaining fold is used for testing.–Results are averaged across folds to compute performance metrics such as F1-score and AUC.

##### Hyperparameter Optimization

Grid search is performed to find the optimal values for the following:–Hidden dimension (hidden_dim).–Learning rate (η).–Number of training epochs (epochs).

The hyperparameter grid is defined as in Equation (Equation 7):(7)$$P=\begin{array}{l} \mathrm{hidden}\_\dim:[32,64],\\ \left\{\mathrm{leaning}\_\mathrm{rate}:[0.001,0.01],\right\}\\ \mathrm{epochs}:[10,20] \end{array}$$

### 3.2. Masking and Feature Extraction

Our proposed segmentation and feature extraction workflow is designed to facilitate accurate segmentation and comprehensive feature extraction for cervical abnormal cell detection. The workflow is divided into four main stages: data annotation, augmentation, segmentation model training, and feature extraction.

We started with a dataset consisting of cervical images (normal: 136, CIN1: 278, CIN2: 286, CIN3: 296, carcinoma: 200). Annotation was performed using LabelMe, and the annotations were saved as .json files [63]. To generate ground truth masks, we developed a custom Python script. This script processes the JSON annotation files, initializes a blank grayscale mask for each image, and iteratively fills the polygonal shapes in the annotations onto the mask. This step ensures precise and accurate mask generation corresponding to the annotated regions in each image.

To increase the diversity of the training dataset, we employed the Albumentations library for data augmentation [64]. Augmentations were uniformly applied to both the denoised (via Non-Local Means) images and their corresponding masks. The transformations included random rotations (90°, 180°, or 270°), horizontal and vertical flipping, brightness and contrast adjustments, and uniform image scaling. This pipeline generated 900 augmented image–mask pairs per class, resulting in a dataset of 4500 samples across five classes.

For segmentation, we employed the DeepLabV3+ architecture [65], which was optimized for pixel-level classification tasks. We chose DeepLabV3+ for the colposcopy images due to its ability to capture fine-grained spatial details and semantic context through the use of Atrous Spatial Pyramid Pooling (ASPP), which effectively handles scale variations in the data. Additionally, its encoder–decoder structure ensures precise boundary detection, making it well suited for the intricate segmentation challenges posed by colposcopy images, where accurate delineation of lesions is critical for diagnosis. The architecture’s proven performance in medical imaging tasks further supports its application in colposcopy image analysis. The training process used the configurations detailed in Table 1. The DeepLabV3+ architecture utilized ResNet50 as its base model and integrated an ASPP (Atrous Spatial Pyramid Pooling) module to capture features at multiple scales. EarlyStopping and ReduceLROnPlateau callbacks were implemented to prevent overfitting and optimize training efficiency. The model’s final upsampling layers resized the output to match the original image dimensions for mask prediction.

After segmentation, the predicted masks were analyzed to extract features for each class. The feature extraction process involved both geometric and texture analysis:Geometric Features: These included area, perimeter, eccentricity, solidity, major and minor axis lengths, aspect ratio, compactness, and circularity.Texture Features: These were derived using the GLCM (Gray-Level Co-Occurrence Matrix) and included contrast, correlation, energy, and homogeneity [66].

The intensity values of the grayscale images were normalized to between 0 and 1 to ensure consistency during feature extraction. The extracted features for each connected region in the predicted masks were saved as structured CSV files for further analysis.

### 3.3. Graph Generation and Augmentation

Our proposed graph-based feature representation workflow transforms segmented image features into graph structures, enabling class-specific analysis and augmentation [22,23]. The workflow consists of two main stages: initial graph generation and augmented graph generation.

In the first stage, we extracted geometric and texture features from segmented masks for five classes (normal, CIN1, CIN2, CIN3, carcinoma), stored them in CSV files, and constructed undirected and fully connected graphs [26]. Nodes represent segmented regions, and edges are based on Euclidean distances between feature vectors, with weights computed as the inverse of distances. Graphs were saved in GML format and visualized using a spring layout.

In the second stage, we expanded the dataset by generating 500 augmented graphs per class. Gaussian noise was applied to feature values to introduce variability while maintaining structural integrity. Edge creation thresholds were randomized between 30% and 60% of the maximum feature space distance, ensuring diverse graph structures. The edge weight formula remained as in Equation (Equation 8):(8)$${\mathrm{Weight}}_{ij}=\frac{1}{{\mathrm{Distance}}_{ij}+\epsilon }$$
where ${\mathrm{Distance}}_{ij}$ is the Euclidean distance between nodes *i* and *j*, and ϵ prevents division by zero.

The entire workflow is summarized in Table 2. This workflow bridges the gap between feature representation and graph-based learning, enhancing cervical abnormal cell detection. Figure 2 presents sample graphs of the five classes where the blue dots in each graphs are the pixels representing the extracted features after segmentation.

### 3.4. GNN (Graph Neural Network) for Final Mapping and Classification with K-Fold Cross-Validation and Hyperparameter Fine-Tuning

#### 3.4.1. Graph Neural Networks (GNNs)

Graph Neural Networks (GNNs) [67] are specialized neural networks designed to work with graph-structured data. Each node in the graph is represented by features, and edges encode relationships between nodes. The GNN aggregates information from neighboring nodes using layers like GCNConv, as represented in Equation (Equation 9):(9)$$h_{i}^{\left(k\right)}=\sigma \left(W^{\left(k\right)}\cdot \mathrm{AGGREGATE}\left(\{h_{j}^{\left(k-1\right)}:j\in N\left(i\right)\}\right)\right)$$
where $h_{i}^{\left(k\right)}$ is node *i*’s embedding at layer *k*, $W^{\left(k\right)}$ are the learnable weights at layer *k*, N(i) are the neighbors of node *i*, and σ is the activation function (e.g., ReLU). In the final layer, a fully connected (FC) layer maps the embeddings to the desired number of output classes.

The specific purpose of the GNN in colposcopy image classification is to leverage graph-based representations of features extracted from colposcopy images. The nodes represent regions of interest (e.g., segmented lesions or cells), and the edges capture spatial or contextual relationships between these regions. By aggregating information across connected regions, the GNN effectively models local and global dependencies in the data, which is critical for distinguishing between different classes (e.g., normal, CIN1, CIN2, CIN3, carcinoma). This approach improves classification performance by capturing the complex spatial and structural patterns unique to colposcopy images.

#### 3.4.2. Graph-to-Mask-to-Image Mapping

The mapping step aligns each graph with its corresponding mask and image. For every graph file, a mask and image are associated, ensuring a direct relationship between graph-based representations, segmented regions, and original images, as represented in Equation (Equation 10):(10)$$\mathrm{Mapping:}\;G_{i}\to \left(M_{i},I_{i}\right)$$
where $G_{i}$ is the graph, $M_{i}$ is the mask, and $I_{i}$ is the image. This mapping ensures that predictions made using the graph can be visualized on the corresponding mask and image.

#### 3.4.3. K-Fold Cross-Validation

K-fold cross-validation splits the dataset into *k* subsets (folds). For each fold, k−1 folds are used for training, and the remaining fold is used for validation or testing. This ensures a robust evaluation and reduces overfitting, as represented in Equation (Equation 11):(11)$$\mathrm{Dataset}=\overset{k}{\underset{i=1}{⋃}}{\mathrm{Fold}}_{i},\;\mathrm{where}\;{\mathrm{Fold}}_{i}\cap {\mathrm{Fold}}_{j}=\emptyset \;\mathrm{for}\;i\neq j$$

For fold *i*, the following apply:Training set: $\mathrm{Dataset}-{\mathrm{Fold}}_{i}$.Test set: ${\mathrm{Fold}}_{i}$.

#### 3.4.4. Hyperparameter Fine-Tuning

Hyperparameter tuning optimizes the model’s performance by searching through a predefined grid of hyperparameters. Each combination is evaluated using validation performance, as represented in Equation (Equation 12):(12)Param Grid={(h,η,e):h∈H,η∈L,e∈E}
where *h* is the hidden dimension, η is the learning rate, *e* are the epochs, and H,L,E are sets of possible values for the parameters. The best combination maximizes a metric, such as the macro F1-score.

#### 3.4.5. Final Classification

After training, the GNN predicts class labels for each graph. The final predictions are evaluated using metrics such as Accuracy, Precision, Recall, F1-score, and the confusion matrix, as represented in Equation (Equation 13):(13)$$\hat{y}=\mathrm{argmax}\left(f_{\theta }\left(G\right)\right),\;f_{\theta }:\mathrm{Graph}\to \mathrm{Class}\;\mathrm{Scores}$$
where $f_{\theta }$ is the GNN model, and $\hat{y}$ is the predicted class. The evaluation metrics used in this study are summarized in the following, reflecting the performance of the model across the K-fold cross-validation and fine-tuning phases, and are explained by Equations (Equation 14)–(Equation 19):(14)$${\mathrm{ATA}}_{n}=\frac{\mathrm{Correct}\;\mathrm{Predictions}\;\mathrm{during}\;\mathrm{Training}}{\mathrm{Total}\;\mathrm{Training}\;\mathrm{Samples}}\times 100$$(15)$${\mathrm{ATL}}_{n}=\frac{1}{N}\sum_{i=1}^{N}\mathrm{Cross}-\mathrm{Entropy}\;\mathrm{Loss}\;\mathrm{during}\;\mathrm{Training}$$(16)$${\mathrm{AVA}}_{n}=\frac{\mathrm{Correct}\;\mathrm{Predictions}\;\mathrm{during}\;\mathrm{Validation}}{\mathrm{Total}\;\mathrm{Validation}\;\mathrm{Samples}}\times 100$$(17)$${\mathrm{AVL}}_{n}=\frac{1}{N}\sum_{i=1}^{N}\mathrm{Cross}-\mathrm{Entropy}\;\mathrm{Loss}\;\mathrm{during}\;\mathrm{Validation}$$(18)$${\mathrm{AP}}_{n}=\frac{\mathrm{TP}}{\mathrm{TP}+\mathrm{FP}},\;{\mathrm{AR}}_{n}=\frac{\mathrm{TP}}{\mathrm{TP}+\mathrm{FN}}$$(19)$$\mathrm{AF}1_{n}=\frac{2\cdot {\mathrm{AP}}_{n}\cdot {\mathrm{AR}}_{n}}{{\mathrm{AP}}_{n}+{\mathrm{AR}}_{n}}$$

In addition to the class-wise metrics, the overall **macro-average F1-score** and **macro-average validation accuracy** provide a holistic evaluation of the model’s performance across all classes, as depicted in Equations (Equation 20) and (Equation 21):(20)$$\mathrm{AF}1\_{\mathrm{Macro}}_{n}=\frac{1}{C}\sum_{c=1}^{C}\mathrm{AF}1_{c,n}$$(21)$$\mathrm{AVA}\_{\mathrm{Macro}}_{n}=\frac{1}{C}\sum_{c=1}^{C}{\mathrm{AVA}}_{c,n}$$
where *C* is the total number of classes, $\mathrm{AF}1_{c,n}$ is the F1-score for class *c*, and ${\mathrm{AVA}}_{c,n}$ is the validation accuracy for class *c* over *n* epochs.

The macro-average metrics are computed during two distinct phases:K-Fold Cross-Validation Phase (1–100 epochs): The macro-average F1-score ($\mathrm{AF}1\_{\mathrm{Macro}}_{100}$) and macro-average validation accuracy ($\mathrm{AVA}\_{\mathrm{Macro}}_{100}$) are computed to provide an overall evaluation of the model’s performance across all classes during the initial training phase.Fine-Tuning Phase (101–151 epochs): The macro-average F1-score ($\mathrm{AF}1\_{\mathrm{Macro}}_{50}$) and macro-average validation accuracy ($\mathrm{AVA}\_{\mathrm{Macro}}_{50}$) are calculated post fine-tuning to reflect the improvement in the model’s classification performance.

The evaluation metrics are computed during these two phases, with subscripts 100 and 50 denoting their respective epochs. Fine-tuning yields significant improvements across all metrics, with early stopping applied when no validation loss improvement is observed for three consecutive epochs.

We employed LIME [68] to interpret model predictions and highlight lesion-specific regions in the colposcopy images. LIME, or Local Interpretable Model-Agnostic Explanations, approximates complex models locally using simple surrogate models to provide intuitive, instance-level explanations. Being model-agnostic, it can be applied across diverse architectures, and is particularly effective for understanding image classification decisions. In our case, LIME enabled visualization of which regions in the cervical image most influenced the model’s prediction, enhancing clinical trust and transparency. Formally, LIME seeks to minimize the objective as shown in Equation (Equation 22):(22)$$\mathrm{explanation}\left(x\right)=L\left(f,g,\Pi_{x}\right)+\Omega \left(g\right)$$
where *f* is the original model, *g* is the interpretable surrogate model, *L* is the locality-aware loss function, $\Pi_{x}$ denotes the proximity measure around instance *x*, and Ω(g) represents the complexity of the explanation model.

This pipeline, depicted in Theorem 1, ensures efficient training and evaluation of the GNN while maintaining interpretability through visualizations.

**Theorem** **1**(Graph Classification via Two-Layer GCN). *Let G=(V,E) be an undirected graph with |V|=n nodes and node feature matrix $X\in R^{n\times F}$. Let $\hat{A}={\tilde{D}}^{-1/2}\tilde{A}{\tilde{D}}^{-1/2}$ denote the symmetrically normalized adjacency matrix with self-loops added, where $\tilde{A}=A+I_{n}$ and $\tilde{D}$ is its degree matrix. A two-layer Graph Convolutional Network (GCN) with weights $W^{\left(1\right)}\in R^{F\times F^{\prime }}$ and $W^{\left(2\right)}\in R^{F^{\prime }\times F^{\prime \prime }}$ maps node features to graph-level logits via*$$\begin{array}{cc} H^{\left(1\right)} & =\sigma (\hat{A}XW^{\left(1\right)}),\\ H^{\left(2\right)} & =\sigma (\hat{A}H^{\left(1\right)}W^{\left(2\right)}),\\ H_{\mathrm{graph}} & =\frac{1}{\left|V\right|}\sum_{v\in V}H_{v}^{\left(2\right)},\\ H_{\mathrm{output}} & =W_{c}H_{\mathrm{graph}}+b_{c}, \end{array}$$*where σ is a non-linear activation (e.g., ReLU), and $W_{c}$ and $b_{c}$ are the fully connected layer’s weights and bias. The predicted class is given by the following:*
$$\hat{y}=\arg\max_{j}\left(\frac{e^{{\left(H_{\mathrm{output}}\right)}_{j}}}{\sum_{k}e^{{\left(H_{\mathrm{output}}\right)}_{k}}}\right).$$

**Proof of Theorem** **1.** The GCN operates in stages:
**Layer 1:** Node features are linearly transformed and aggregated using the normalized adjacency matrix to produce embeddings $H^{\left(1\right)}$.**Layer 2:** The embeddings are further transformed and aggregated to produce $H^{\left(2\right)}$.**Pooling:** Node embeddings $H^{\left(2\right)}$ are aggregated (mean pooling) into a graph-level embedding $H_{\mathrm{graph}}$.**Classification:** A fully connected layer maps $H_{\mathrm{graph}}$ to class logits $H_{\mathrm{output}}$, which are passed through softmax to obtain class probabilities. The final predicted label $\hat{y}$ is the class with the highest probability.
□

### 3.5. Datasets and Handling of Class Imbalance

We used four datasets in this study, including a primary colposcopy image dataset of 1200 patients with an average age of 27 years received personally from IARC [69], where the dataset was supported by metadata comprising 1180 entries in a .csv file. Regarding diagnoses, 11.33% of the cases were normal, 23.17% were CIN1, 23.83% were CIN2, 24.67% were CIN3, and 16.67% were carcinoma. The images were obtained in Lugol’s iodine (7% of the images), acetic acid (89% of the images), and normal saline (4% of the images). A secondary colposcopy dataset from Kaggle called Malhari containing 2790 images of three classes (CIN1, CIN2, and CIN3) [70], a secondary eye_disease dataset from Kaggle [71] containing 4217 images of four classes (normal, cataract, diabetic retinopathy, and glaucoma), and a secondary brain tumor MRI dataset from Kaggle containing 7022 images of four classes (no tumor, glioma, meningioma, and pituitary) [72] were used in addition to the primary dataset. The primary dataset originally had a high percentage of class imbalance, which we handled by adding data as described in Section 3.2. The Malhari dataset had images captured in three different solutions: Lugol’s iodine, acetic acid, and normal saline. In the original dataset, CIN1, CIN2, and CIN3 were distributed as follows: 900, 930, and 960, respectively. The overall class imbalance ratio (IR) was 1.07, indicating a fairly balanced dataset with only a slight imbalance among the classes. Hence, we did not apply data augmentation to this dataset but it was denoised using NLM. The eye_disease dataset contains 1038 cataract images, 1098 diabetic_retinopathy images, 1007 glaucoma images, and 1074 normal images and has a class imbalance ratio (IR) of 1.09, indicating a mild class imbalance. Hence, we did not apply data augmentation to this dataset but it was denoised using NLM. The actual brain MRI dataset contains 7022 images, with some files missing from the original dataset. The distribution includes 1595 no tumor, 1457 pituitary, 1339 meningioma, and 1321 glioma images. The overall class imbalance ratio (IR) for the given dataset is 1.21, indicating a moderate imbalance, with the “no tumor” class being the most represented and “glioma” the least. To ensure optimal quality, we applied a denoising process to enhance image clarity while preserving critical diagnostic features. However, no augmentation was performed on this dataset, to maintain its original integrity for model evaluation. The primary dataset was used for our study, and the three secondary datasets were used to test the generalization of the model. The three secondary datasets are licensed under the public domain. The dimension of the primary dataset is 800×600, the dimension of the Malhari dataset is 640×480, the dimension of the eye_disease dataset is 256×256, and the dimension of the brain MRI dataset is 256×256, and all were resized to 128×128. The results of the augmentation applied to the primary dataset and sample images of Malhari and brain MRI dataset are shown in Figure 3. The size of the sample on which the entire experiment was performed is shown in Table 3.

### 3.6. Data Availability and Ethical Considerations

The datasets utilized in this study were obtained from a variety of sources and are subject to different licensing terms. We ensured that all datasets were used responsibly, adhering to the respective licensing conditions and maintaining compliance with ethical research standards. A detailed overview of the licensing and usage conditions for each dataset is provided below:IARC [69]: This colposcopy dataset was personally provided to the author via email communication by the respective custodians of the IARC colposcopy data bank. The dataset includes separate metadata files in .csv format. It was utilized in this research solely for academic purposes. As the dataset was shared directly for research use, and no specific licensing terms were provided, it was used in accordance with standard academic research practices.Malhari [70]: This dataset is publicly available on the Kaggle platform; however, no explicit licensing information was provided by the dataset uploader. In accordance with best academic practices, the dataset was used solely for non-commercial academic research purposes. No modifications were made to the original data, and no redistribution of the dataset was performed. Furthermore, no patient-identifiable information was included or processed. This use complies with fair research practices for publicly available datasets lacking explicit licensing terms.eye_disease dataset [71]: The secondary dataset was obtained under the Open Data Commons Open Database License (ODbL) v1.0. In accordance with the license, proper attribution has been provided, and the dataset was used exclusively for non-commercial academic research purposes.Brain tumor MRI dataset [72]: This dataset is available under the Creative Commons CC0 1.0 Universal (Public Domain Dedication) license. It can be freely used, modified, and distributed without restriction. In this study, the dataset was utilized strictly for academic research purposes. Although attribution is not legally required under CC0, appropriate credit has been provided to acknowledge the source.

## 4. Results

### 4.1. Evaluation Metrics and Implementation Details

The evaluation of both segmentation and GCN (Graph Convolutional Network) models involved multiple metrics and specific execution environments. Below is a detailed description of these aspects.

#### 4.1.1. Metrics

For the segmentation model, the evaluation metrics included the Dice Coefficient, Intersection over Union (IoU), Precision, Recall, and the Binary Cross-Entropy (BCE) loss function. The formulas for these metrics are as shown in Equations (Equation 23)–(Equation 27):Dice Coefficient:(23)$$\mathrm{Dice}=\frac{2\times TP}{2\times TP+FP+FN}$$Intersection over Union (IoU):(24)$$\mathrm{IoU}=\frac{TP}{TP+FP+FN}$$Precision:(25)$$\mathrm{Precision}=\frac{TP}{TP+FP}$$Recall:(26)$$\mathrm{Recall}=\frac{TP}{TP+FN}$$Binary Cross-Entropy Loss:(27)$$\mathrm{BCE}=-\frac{1}{N}\sum_{i=1}^{N}\left[y_{i}\log\left({\hat{y}}_{i}\right)+\left(1-y_{i}\right)\log\left(1-{\hat{y}}_{i}\right)\right]$$

For the GCN model, the evaluation metrics are as illustrated in Section 3.4.5.

#### 4.1.2. Training Details

For the segmentation model, training was conducted over 100 epochs with a batch size of 8.

Similarly, the GCN model underwent an initial training phase of 100 epochs, followed by an additional 100 epochs of fine-tuning. A batch size of 8 was maintained throughout the training process.

To ensure robustness, a five-fold cross-validation (K=5) strategy was employed. Since k_folds=5, the dataset was divided as follows:Training Set: 80% of the data four folds).Validation Set: A subset of the training set, used for hyperparameter tuning (grid search).Test Set: 20% of the data one fold), used for final evaluation.

#### 4.1.3. Hyperparameter Tuning

Hyperparameter tuning was not applicable to the segmentation model. However, for the GCN model, hyperparameters were optimized using the following values:Hidden Dimension: [32, 64];Learning Rate: [0.001, 0.01];Epochs: 100;Early Stopping: Yes.

#### 4.1.4. Execution Environment

The models were trained and evaluated in the following computational environment:Operating System: Windows 10 Home Single Language;Python Version: 3.10.12 (main, 6 November 2024, 20:22:13) [GCC 11.4.0];TensorFlow Version: 2.15.0;Keras Version: 2.15.0;Hardware Accelerator: Colab Pro, v2-8 TPU;CPU: Intel(R) Core(TM) i5-1035G1 CPU @ 1.00 GHz, 1.19 GHz;Installed RAM: 16.0 GB (15.8 GB usable).

### 4.2. Result of Execution of the Proposed Model on the Primary Augmented Dataset

#### 4.2.1. Inference Time per Image, Computational Complexity, and Real-Time Feasibility

The inference times for DeepLabV3 and the GCN across different hardware show that both models meet real-time requirements on the TPU (30 ms for DeepLabV3, 20 ms for GCN) and GPU (50 ms for DeepLabV3, 30 ms for GCN). However, on the CPU (400 ms for DeepLabV3, 250 ms for GCN), the models fail to meet real-time constraints, making the TPU and GPU the preferred choices for real-time applications.

The time complexity of DeepLabV3 is $O(N^{2})$, as convolutional layers scale quadratically with the number of pixels, requiring full-image processing for segmentation. In contrast, the GCN has a time complexity of O(V+E), where each node aggregates features from its neighbors, making it linear in relation to the number of nodes and edges.

#### 4.2.2. FLOPs and TPU and GPU Memory Usage During Training

To further evaluate the computational efficiency, we modified the original code to calculate the Floating Point Operations per Second (FLOPs) and peak GPU memory consumption during training. DeepLabV3 requires approximately 40 GFLOPs per image during forward and backward passes, while the GCN requires around 2.5 GFLOPs per graph sample. Peak GPU memory usage during training was observed to be approximately 5.8 GB for DeepLabV3 and 2.1 GB for GCN on an NVIDIA V100 GPU with 32 GB VRAM. The inference times measured on TPU, GPU, and CPU platforms are summarized in Table 4, confirming that both models are computationally efficient and suitable for deployment on real-time medical imaging platforms.

#### 4.2.3. Masking

##### Qualitative Results

The qualitative results of our segmentation approach, illustrated in Figure 4, depict the carcinoma class in three datasets. As is evident from the images, the predicted masks closely resemble the ground truth masks in shape and structure across all three datasets, demonstrating the effectiveness of our approach.

##### Quantitative Results

The quantitative results of our segmentation approach, presented in this section, pertain exclusively to our primary dataset. According to the segmentation metrics detailed in Section 4.1, we provide the metric results on a per-class basis, along with the class-wise plot of cross-entropy loss (training and validation). Table 5 summarizes the metrics obtained for the five classes, while Figure 5 illustrates the class-wise loss plot captured for every 20 epochs from epoch 0 to epoch 100. This table presents segmentation metrics obtained from experiments conducted on 900 colposcopy images and 900 masks per class over 100 epochs with a batch size of 8 utilizing the DeepLabV3+ architecture. The ground truth masks were generated using a Python script that processed .json files containing annotation details.

#### 4.2.4. Results of Feature Extraction and Analysis

Based on the feature extraction method and the extracted features outlined in Section 3.2, we present a comprehensive comparison and analysis of the features across the five classes.

The morphological and texture-based features reveal distinct variations among the five classes, as summarized in Table 6. The normaclass exhibits a broad distribution of area and moderate contrast with high circularity values. In contrast, CIN1 shows reduced variability in area and contrast, while circularity becomes more skewed. CIN2 and CIN3 display a progressive increase in contrast spread and a decrease in circularity, indicating irregular structural changes. Carcinoma exhibits the most pronounced deviation, with highly variable area, extreme contrast variations, and significantly lower circularity, highlighting the aggressive morphological alterations associated with malignancy.

#### 4.2.5. Results and Analysis of the Proposed Graph Neural Network on the Primary Dataset

The proposed Graph Neural Network (GNN), executed within the environment detailed in Section 4.1, was initially trained for 100 epochs using a batch size of 8 on a dataset comprising 500 graphs in five classes. The model architecture includes two graph convolutional layers, followed by global mean pooling and a fully connected classification layer. Training is performed using cross-entropy loss with batch-wise optimization. A *K*-fold cross-validation approach (K=5) is used, ensuring robust evaluation by partitioning the data into multiple training and validation folds. Furthermore, hyperparameter tuning is performed through grid search, optimizing hidden_dim (32, 64) and learning rates (0.001, 0.01) across multiple epochs.

##### Qualitative Results in the Validation Dataset

This section illustrates the uniqueness of the proposed architecture. The function generate_graph_to_mask_and_image_mapping establishes a correspondence between graph representations, segmentation masks, and colposcopy images, ensuring proper alignment of these modalities. This mapping facilitates multi-modal learning by integrating structural and spatial information, where graphs capture the topological structure of tissue regions derived from segmentation masks. The Graph Convolutional Network (GCN) leverages this mapping to process relational dependencies within the image, enhancing feature extraction and class discrimination. By associating graphs with their corresponding images and masks, the architecture ensures consistency in data organization, leading to improved learning efficiency and better classification performance in cervical lesion analysis. Figure 6 shows the mapping of the class-wise graph, predicted mask, original colposcopy image, and predicted colposcopy image of the test cases. It also illustrates the segmentation and graph representation pipeline, where the original colposcopy images undergo lesion segmentation, and the predicted masks are compared against ground truth masks. The overlay images confirm the effectiveness of segmentation, while the final graph representations capture lesion structures, which are useful for classification. The predicted masks closely resemble the ground truth, indicating a reliable segmentation model.

Figure 7 presents a sample prediction from the test set, consisting of 900 images, generated using the proposed algorithm. The figure illustrates both the true and predicted classes, as verified by Dr. Srikanth. Figure 8 presents the ROC-AUC curve for the five classes, while Figure 9 illustrates the corresponding confusion matrix. Both of these visualizations are for the test samples (900 images) in our case.

Figure 8 presents the ROC-AUC curves for the five-class classification task, demonstrating high model performance with AUC values ranging between 0.88 and 0.99. The carcinoma and augmented normal classes achieve near-perfect discrimination (AUC = 0.99), whereas CIN1 and CIN2 have slightly lower AUC values (0.88–0.90), indicating some feature overlap between these categories.

Figure 9 shows the confusion matrix for the test set, where strong diagonal dominance suggests high classification accuracy. The augmented normal class has nearly perfect predictions, aligning with its high AUC. However, some misclassifications occur between CIN1 and CIN2, as well as between CIN3 and carcinoma, indicating a degree of feature similarity between these conditions. This highlights areas where further refinement in feature extraction or model tuning may be beneficial. Overall, the model demonstrates strong classification performance, with minor challenges in differentiating closely related CIN stages.

##### Analysis of False Positives and False Negatives

Based on the confusion matrix (Figure 9), the class-wise false positives (FP) and false negatives (FN) were analyzed. For CIN1, 33 false negatives and 33 false positives were observed, mostly involving confusion with CIN2 and CIN3. For CIN2, 27 false negatives and 32 false positives were recorded. The normal class showed 25 false negatives and 21 false positives, indicating reliable detection of non-pathological cases. Carcinoma detection exhibited only 30 false negatives and 24 false positives, highlighting the model’s high sensitivity and specificity for malignant cases. CIN3 exhibited 23 false negatives and 28 false positives. These results demonstrate the model’s effectiveness in accurately classifying critical lesion classes while minimizing diagnostic errors.

##### Quantitative Results in the Validation Dataset

Table 7 summarizes the class-wise performance metrics for the model across two distinct phases: before fine-tuning ($M_{100}$) and after fine-tuning ($M_{50}$).

#### 4.2.6. Proposed Model Explainability by LIME

Based on the LIME visualizations and the corresponding quantitative evaluation in Table 8, several key insights can be drawn regarding the interpretability and reliability of the proposed classification model for colposcopy images.

The LIME-generated images in Figure 10 clearly highlight the discriminative lesion regions contributing to the model’s predictions across all five classes: normal, CIN1, CIN2, CIN3, and carcinoma. The highlighted regions (in yellow and red) are consistent with pathological patterns observed in clinical diagnosis, indicating that the model focuses on medically relevant features during decision-making. In the LIME visualizations, red and yellow areas represent regions that contribute most strongly to the model’s prediction, while green areas indicate regions with less or negative contribution. Some incomplete regions in the visualization are due to cropping during overlay generation and do not affect the interpretability of the discriminative lesion areas. This improves the overall clinical trustworthiness of the system.

Quantitatively, Table 8 presents the class-wise evaluation of LIME using Fidelity Score, Stability Score, and Local Accuracy. Among the five classes, carcinomaachieved the highest Fidelity Score (92.1%) and Local Accuracy (94.2%), indicating the strongest correlation between the model’s confidence and the generated explanations for this class. CIN3 also performed consistently well across all three metrics, reflecting the model’s interpretability in higher-grade lesion detection.

Although the normal and CIN1 classes recorded slightly lower scores, their Fidelity and Stability values remained above 87%, suggesting that the generated explanations were still consistent and interpretable, albeit with slightly less clarity due to subtler lesion boundaries.

On average, the model achieved a Fidelity Score of 91.0%, Stability Score of 89.1%, and Local Accuracy of 92.8%. These high values confirm that LIME effectively aligns with the model’s decision process and remains stable under perturbations. This reinforces LIME’s potential as a reliable post hoc explanation technique for validating classification outcomes in colposcopy-based cervical lesion diagnosis.

##### Misclassifications by the Model of the Brain MRI Dataset and Eye_Disease Dataset Explained by LIME

In both the brain tumor and eye disease (EE) classification models, LIME was used to visualize and understand the model’s misclassifications. For the brain dataset, LIME highlighted that misclassifications often occurred due to overlapping visual patterns across tumor types, with attention focused on non-tumor regions or image edges. In the eye dataset, misclassifications mainly happened between visually similar conditions like cataract and normal, where LIME showed the model emphasizing irrelevant background or poorly illuminated areas. These insights help in identifying dataset limitations and improving model robustness through better preprocessing or targeted augmentation, as illustrated in Figure 11.

#### 4.2.7. Ablation Studies

##### Model Generalization on Brain Tumor MRI Dataset, the Malhari Dataset and the Eye_Disease Dataset Without Augmentation

Qualitative Evaluation

We used the secondary datasets, the brain MRI, eye_disease, and Malhari datasets, as illustrated in Section 3.5, to understand the behavior of the model on datasets without augmentation. Figure 12, Figure 13 and Figure 14 present the classification predictions based on the true labels of the test dataset of the brain MRI, Malhari, and eye_disease datasets, respectively.

Quantitative Evaluation

Table 9 summarizes the segmentation and classification performance for the Malhari colposcopy, brain MRI, and eye disease datasets. In the Malhari dataset, segmentation performance varied across staining solutions, with Lugol’s iodine achieving the highest Dice Coefficient (86.1%) and IoU (85.6%) for CIN3, while normal saline had the lowest values, indicating the influence of staining on model performance. Classification results improved after fine-tuning, with the macro-average F1-score increasing from 89.4% to 90.56% and validation accuracy increasing from 89.88% to 91.02%, demonstrating the effectiveness of augmentation in mitigating minor class imbalance (IR = 1.07).

For the brain MRI dataset, segmentation performance was generally higher, with pituitary tumors achieving the best Dice Coefficient (92.56%) and IoU (86.1%), while no tumor had the lowest IoU (82.1%) due to challenges in distinguishing normal tissue. Classification fine-tuning resulted in a more significant improvement compared to the Malhari dataset, with the macro-average F1-score rising from 88.4% to 91.56% and validation accuracy rising from 90.1% to 91.98%. The greater gains in the brain MRI dataset suggest that well-separated tumor classes and reduced class imbalance contribute to better generalization after augmentation. Fine-tuning and early stopping effectively optimized classification in both datasets, confirming the necessity of tailored augmentation strategies for different medical imaging modalities.

In the eye disease dataset, segmentation results were highly consistent, with glaucoma achieving the highest Dice Coefficient (93.56%) and IoU (92.1%). The classification performance also showed notable improvement after fine-tuning, with macro-average F1-score increasing from 89.4% to 90.56% and validation accuracy rising from 92.1% to 94.98%. The class imbalance ratio of 1.09 reflects a relatively balanced dataset, which likely contributed to the model’s strong performance. These results emphasize the generalizability of the proposed method across varied modalities, including retinal imaging, and reinforce the impact of fine-tuning and early stopping in achieving robust classification performance.

##### Comparative Analysis of Fixed Graph Weights, Focal Loss, and the Proposed Augmented Adaptive Graph Learning Framework

Comparative Analysis of Graph Learning Strategies.

To evaluate the effectiveness of our proposed adaptive graph learning framework, we compared it with two baseline strategies: (1) fixed-weight graph construction based on handcrafted features, and (2) focal loss training without augmentation.

In the fixed-weight graph strategy, edge weights were statically assigned using the Euclidean distance between nodes based on morphological features (e.g., area, perimeter, eccentricity) and texture descriptors (e.g., GLCM contrast, energy, homogeneity). To avoid overly dense graphs, we employed threshold-based pruning to retain only the most similar node connections. Although this method preserved the basic relational structure, it lacked the ability to adapt weights during training, limiting its ability to capture higher-order contextual dependencies. In contrast, our proposed model dynamically learns edge weights through end-to-end optimization, allowing the graph topology to evolve during training, thereby improving feature propagation and class separability.

In the second baseline, focal loss was used to mitigate class imbalance by down-weighting well-classified examples and focusing on hard instances. However, this approach did not increase the diversity of the data. Our model integrates class-balanced data augmentation and early stopping, which improved both convergence speed and generalization capability.

As shown in Table 10, the proposed model consistently outperforms both baselines in all metrics, including macro-average F1-score and validation accuracy, while requiring fewer training epochs due to faster convergence.

Comparison with the baseline models on our primary dataset.

Table 11 shows the performance comparison of the proposed architecture with EfficientNet, ViT, and ResNet50. Metric M1 is the macro-average F1-score ($\mathrm{AF}1\_{\mathrm{Macro}}_{100},\mathrm{AF}1\_{\mathrm{Macro}}_{50}$) and metric M2 is the macro-average validation accuracy ($\mathrm{AVA}\_{\mathrm{Macro}}_{100},\mathrm{AVA}\_{\mathrm{Macro}}_{50}$) for the five classes. As we employed early stopping in our proposed architecture, we ran the baseline models for 50 epochs in the fine-tuning phase. The table presents a comparison of the classification performance before and after fine-tuning on different baseline models (EfficientNet, ViT, and ResNet50) and the proposed model using the primary dataset (Section 3.5). EfficientNet [73], ViT [74], and ResNet50 [73] were selected as baseline models due to their strong performance in image classification tasks. EfficientNet was chosen for its scalability and efficiency, balancing accuracy and computational cost. ViT (Vision Transformer) was included for its ability to capture long-range dependencies in images using self-attention mechanisms. ResNet50, a widely used deep CNN, was selected for its robust feature extraction capabilities and residual connections that mitigate vanishing gradient issues. These models provide a diverse comparison against the proposed approach. The proposed model consistently outperforms all baselines in both M1 and M2 metrics, demonstrating the effectiveness of incorporating segmentation, feature extraction, and graph-based classification. Notably, ResNet50 shows the lowest performance, while ViT performs better than ResNet50 but slightly worse than EfficientNet. The proposed model achieves the highest F1-score and validation accuracy post fine-tuning, validating its robustness. Additionally, early stopping was applied only in the proposed model, ensuring optimized training and preventing overfitting.

Comparison with the known models on our primary dataset.

Table 12 presents a comparative analysis of our proposed approach against various alternative models for our primary dataset, structured as an ablation study. MTANet [75], the Deep Learning Model for Cervical Cancer Prediction [76], and CerviFusionNet [77] are considered as direct classification approaches for our dataset, providing a baseline for understanding how end-to-end deep learning methods perform in comparison to our multi-stage framework. The Segment Anything Model (SAM) [78] replaces DeepLabV3 in our segmentation stage, allowing us to assess whether general-purpose segmentation models can achieve similar performance to task-specific ones. Similarly, Hybrid Deep Feature Extraction [79] is used instead of our dedicated feature extraction module to determine the impact of handcrafted feature selection versus learned representations. For the graph-based classification stage, the **K-Nearest Neighbors (KNN) [8] Graph**, **Graph Attention Networks (GAT) [80]**, and **Graph Transformer Networks (GTN) [81]** serve as replacements for our **Graph Convolutional Network (GCN)**, enabling an evaluation of different graph learning strategies in medical classification tasks. The KNN Graph is included to assess how local neighborhood connections affect classification performance. GAT is chosen for its ability to dynamically assign attention-based weights to edges, improving feature propagation. GTN is selected for its capability to learn hierarchical relationships in graphs, which makes it suitable for complex medical image analysis. These models were chosen due to their established effectiveness in various related domains, and comparing them provides deeper insights into the impact of different graph learning approaches in medical classification tasks. MTANet was selected because it integrates segmentation and classification into a unified attention-based framework, similar to our staged pipeline. The Deep Learning Model for Cervical Cancer Prediction employs transfer learning on multi-modal medical images, making it relevant for comparison with our graph-based approach. CerviFusionNet was included as it combines a CNN and Vision Transformers for colposcopy-based lesion classification, allowing us to compare our graph-based learning approach with their hybrid fusion method. The SAM was used in place of DeepLabV3 to determine whether a state-of-the-art foundation model trained on large-scale data could generalize well to medical segmentation. Hybrid Deep Feature Extraction was introduced to evaluate whether combining deep and handcrafted features yields better performance than purely learned representations. GAT and GTN were incorporated as alternative graph-based learning techniques to examine how attention and transformer mechanisms affect colposcopy image classification. The results indicate that, while direct classification methods such as CerviFusionNet and MTANet achieve competitive performance, they do not surpass our integrated pipeline. The SAM as a segmentation alternative exhibits strong generalization but is marginally less effective in producing segmentation masks optimized for classification. Hybrid Deep Feature Extraction enhances feature learning but does not match the structured representation obtained from our pipeline. Among graph-based models, both GAT and GTN perform well, but GCN maintains a slight edge in leveraging structured colposcopy image data for classification. Overall, our proposed approach achieves the highest scores in both macro-average F1-score (M1) and macro-average validation accuracy (M2), demonstrating that our multi-stage integration of segmentation, feature extraction, and graph-based classification yields superior performance. This study confirms that each component of our framework contributes meaningfully to overall classification accuracy, justifying its selection over alternative methodologies.

Performance of the model on the primary dataset with noise and with less brightness and contrast.

To assess the robustness of the proposed model under real-world perturbations, we conducted an ablation study using synthetically augmented test sets. Specifically, we introduced Gaussian noise and adjusted brightness and contrast to simulate noisy and low-visibility scenarios, respectively. Noise was added by injecting normally distributed perturbations into the image pixel values, while contrast and brightness were altered using a linear transformation. The augmented datasets—Noisyand Low Contrast—were then evaluated using the already fine-tuned model without retraining.

As shown in Table 7, the model exhibited a graceful degradation in performance under perturbations. For noisy images, the macro-average F1-score dropped from 94.56% to 90.6%, and the validation accuracy from 98.98% to 93.6%. In the case of reduced contrast and brightness, the F1-score further declined to 89.2%, and accuracy to 91.6%. Despite these variations, the model maintained high interpretability and generalization capability, reinforcing its suitability for real-world clinical deployment.

## 5. Discussion

The inference performance of DeepLabV3 and GCN across different hardware platforms highlights the practical considerations for real-time deployment. Both models achieve real-time speeds on the TPU and GPU, with GCN outperforming DeepLabV3 in latency—especially on the TPU (20 ms vs. 30 ms) and GPU (30 ms vs. 50 ms). However, on the CPU, both models exhibit significant delays (400 ms for DeepLabV3 and 250 ms for GCN), making them unsuitable for real-time applications in resource-constrained environments. From a computational complexity standpoint, DeepLabV3’s $O(N^{2})$ scaling—due to dense convolutional operations—makes it less efficient for high-resolution inputs, whereas the GCN’s O(V+E) complexity enables more scalable performance as it operates on graph structures. This distinction underscores the GCN’s advantage in environments requiring low latency and efficient computation.

The segmentation metrics for colposcopy images using DeepLabV3+ demonstrate strong performance across all five classes, with Dice Coefficients ranging from 90.5% for the normal class to 93.01% for CIN3 and Precision values consistently above 91%. These results highlight the model’s effectiveness in accurately segmenting different classes, as reflected in the high IoU and Recall values, ensuring robust boundary delineation and minimal false negatives. The loss plots (Figure 5) reveal consistent convergence across classes, with both training and validation losses decreasing steadily, indicating effective learning and minimal overfitting. This combination of metrics and loss trends underscores the model’s reliability and suitability for colposcopy image analysis.

As per Table 7, we can see there is a significant improvements in all metrics following the fine-tuning phase. For instance, training accuracy and validation accuracy demonstrate consistent improvement across all classes, with normal class validation accuracy increasing from 92.3% to 99.1%, and carcinoma class validation accuracy improving from 92.4% to 99.2%. Similarly, the macro-average F1-score and validation accuracy improved substantially, reflecting the model’s ability to generalize better across all classes post optimization. The reduction in training and validation loss across all classes further highlights the effectiveness of fine-tuning in minimizing misclassifications. Precision, Recall, and F1-score metrics also exhibit noticeable enhancements, particularly in more challenging classes such as CIN1 and CIN3, confirming the model’s robustness. Moreover, the application of early stopping during fine-tuning ensures computational efficiency by halting training when no improvement in validation loss is observed for three consecutive epochs. Overall, the table demonstrates the efficacy of the proposed GNN framework in achieving high accuracy and reliability for cervical lesion classification.

## 6. Limitations and Future Research

Despite its effectiveness, our method has some limitations. First, the segmentation stage heavily depends on the quality of DeepLabV3 masks, which may introduce errors if misclassified regions are not refined. Second, the graph construction process can be computationally expensive, particularly for large-scale datasets. Third, our model, while robust, requires extensive hyperparameter tuning to generalize well across different colposcopy datasets.

For future work, we plan to explore self-supervised learning techniques to reduce reliance on manually labeled segmentation data. Additionally, integrating multi-modal information, such as patient history and other clinical biomarkers, could further improve classification accuracy. Finally, optimizing the graph construction step with adaptive neighborhood selection could enhance efficiency while preserving meaningful relationships in medical images.

## 7. Conclusions

In this paper, we proposed a novel multi-stage framework for colposcopy image classification, integrating segmentation using DeepLabV3, feature extraction, graph construction, and classification using a Graph Convolutional Network (GCN). Our approach effectively leverages structured feature representations to enhance classification performance, achieving superior results compared to alternative segmentation and classification methods. The ablation study demonstrated that each component contributes significantly to overall accuracy, highlighting the benefits of graph-based learning in medical image analysis.

Our proposed framework addresses the critical limitations observed in existing graph-based and multi-modal methods by integrating dynamic graph construction with segmentation-based node embeddings. It optimizes feature learning through an adaptive pooling mechanism and ensures robust model generalizability via K-fold cross-validation and fine-tuned hyperparameter selection. By leveraging multi-modal data fusion and improving computational efficiency, the proposed model enhances classification accuracy, scalability, interpretability, and generalization across diverse medical imaging datasets, setting a new benchmark for medical image analysis.

### 7.1. Clinical Application

The model was tested in Apollo Hospitals, Banjara Hills, Hyderabad, in a GPU environment on 1000 colposcopy images containing 256 CIN1 images, 395 CIN2 images, and 349 CIN3 images and we achieved a macro-average validation accuracy of 91.09% and a macro-average F1-score of 90.09%. The segmentation and the classification results were verified by the oncologists in the hospital.

### 7.2. Analysis of Performance on Real-World Clinical Data

The real-world clinical evaluation conducted at Apollo Hospitals achieved a macro-average validation accuracy of 91.09% and a macro-average F1 score of 90.09%. The observed 9% reduction in accuracy compared to the primary dataset is primarily attributable to domain shift (i.e., differences in imaging conditions), variability in oncologist annotations, and the inherent difficulty in distinguishing between intermediate CIN grades. Additionally, we evaluated the model using the original dataset without data augmentation but incorporating focal loss. As shown in Table 10, focal loss alone did not yield comparable improvements in macro-average validation accuracy or F1-score relative to the improvements observed with data augmentation. Notably, high-grade lesions such as CIN3 lesions were still detected with high sensitivity, underscoring the clinical applicability and robustness of the system. Future research will focus on mitigating domain shift through domain adaptation techniques and expanding the dataset with more diverse clinical samples to further enhance model generalization.

## Figures and Tables

**Figure 1 cancers-17-01521-f001:**
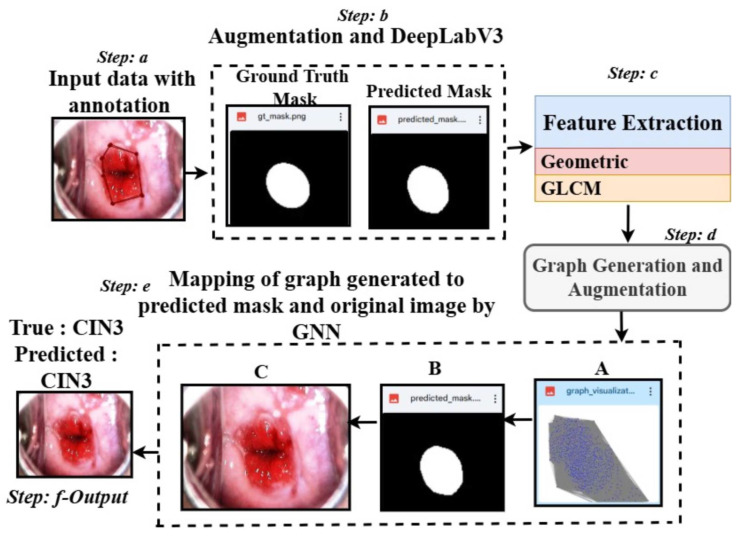
Overview of the proposed architecture: Step a shows a sample CIN3 image with annotation using LabelMe. Step b applies a masking approach using DeepLabV3 and is followed by Step c, where features are extracted using the predicted masks. Step d generates the graph from the extracted features. Step e maps the graph (**A**), the predicted mask (**B**), and the original image (**C**), leading to Step f, which classifies the data into classes.

**Figure 2 cancers-17-01521-f002:**
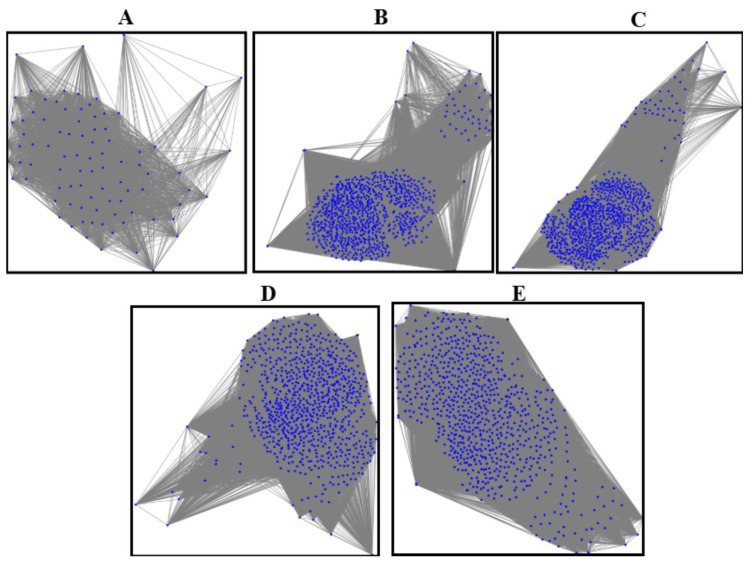
The graphs of five classes: (**A**) graph of the normal class, (**B**) graph of CIN1, (**C**) graph of CIN2, (**D**) graph of CIN3, (**E**) graph of carcinoma.

**Figure 3 cancers-17-01521-f003:**
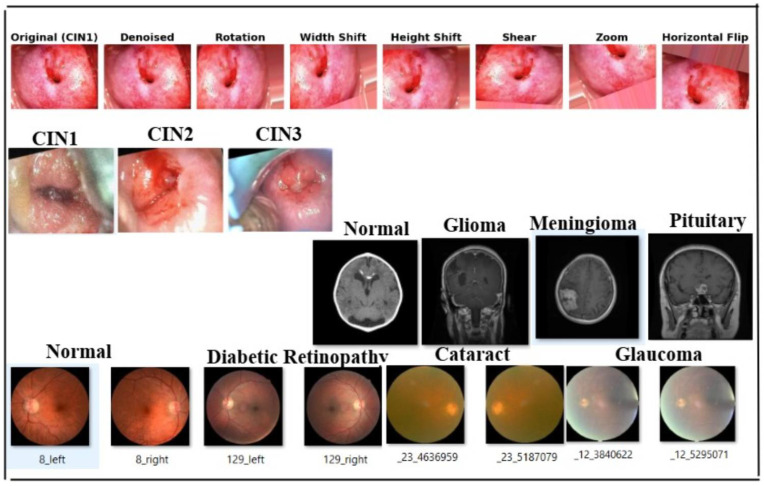
Sample image of four datasets. The first row depicts images of the denoising and augmentation of the primary colposcopy dataset. The second row depicts images of the Malhari secondary denoised dataset without augmentation, the third row depicts image of the brain MRI denoised dataset without augmentation, and the fourth row depicts the eye_disease dataset denoised and without augmentation.

**Figure 4 cancers-17-01521-f004:**
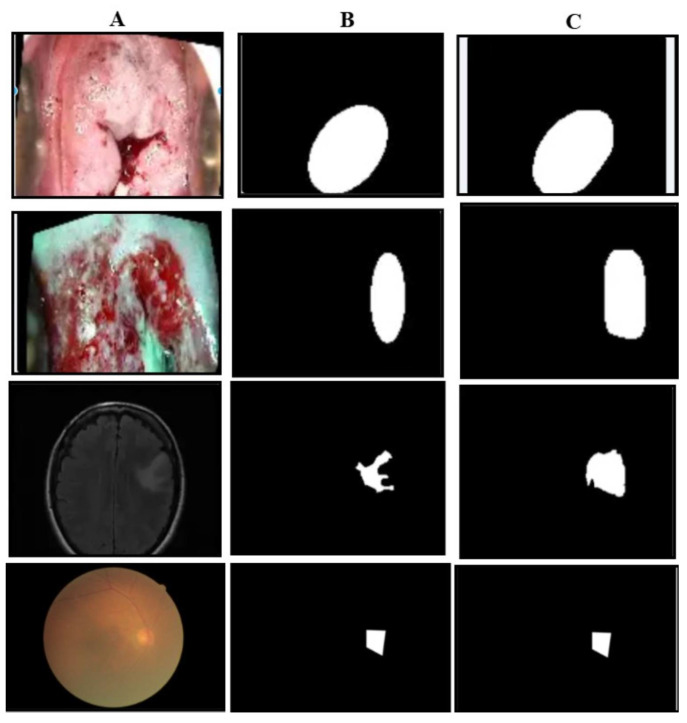
Illustration of masking results: the first row is of carcinoma images of the primary dataset, the second row is the carcinoma images of the Malhari dataset, the third row is the pituitary images of the brain MRI dataset. (**A**) represents the original image, (**B**) represents the ground truth mask, and (**C**) represents the predicted mask.

**Figure 5 cancers-17-01521-f005:**
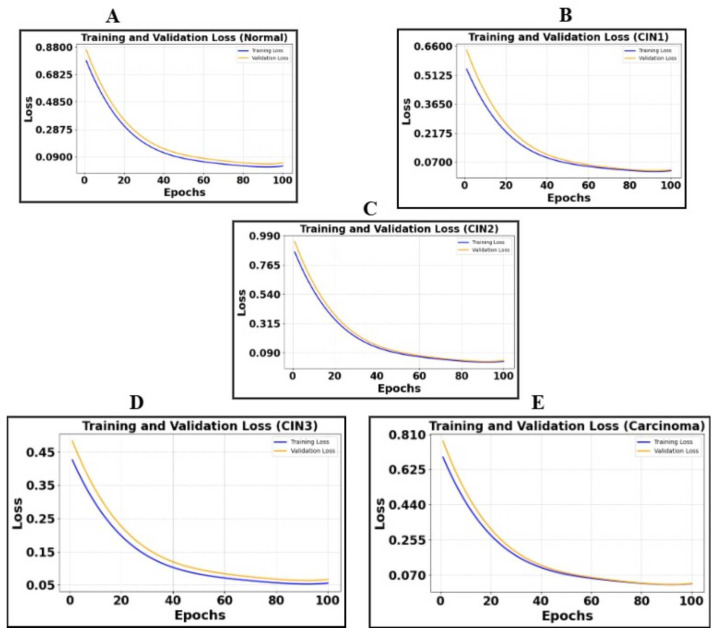
Illustration of cross-entropy plots of five classes. (**A**) is the loss plot of the normal class, (**B**) is the loss plot of CIN1, (**C**) is the loss plot of CIN2, (**D**) is the loss plot of CIN3, and (**E**) is the loss plot of carcinoma.

**Figure 6 cancers-17-01521-f006:**
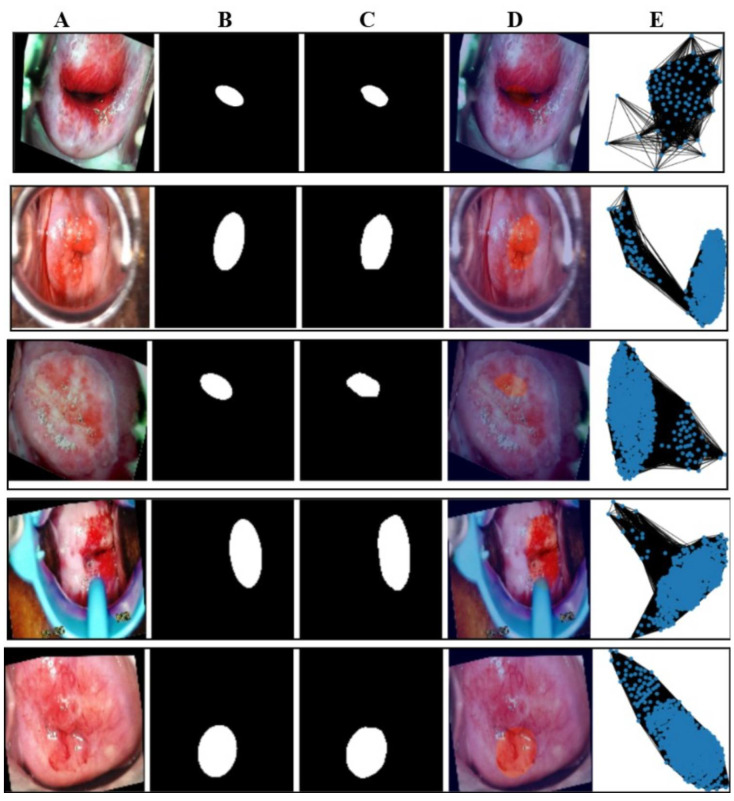
Illustration of the mapping of five classes of the test image. The first row is of the normal class, the second row is of CIN1, the third row is of CIN2, the fourth row is of CIN3, and the fifth row is of carcinoma. The image marked (**A**) in all the rows is the mapping of the original image class, the image marked (**B**) is of the ground truth mask, the image marked as (**C**) is of the predicted mask, the image marked as (**D**) is of the overlay where the lesion is marked as a deep red circle, and the last column (**E**) is the graph generated of the images of column A.

**Figure 7 cancers-17-01521-f007:**
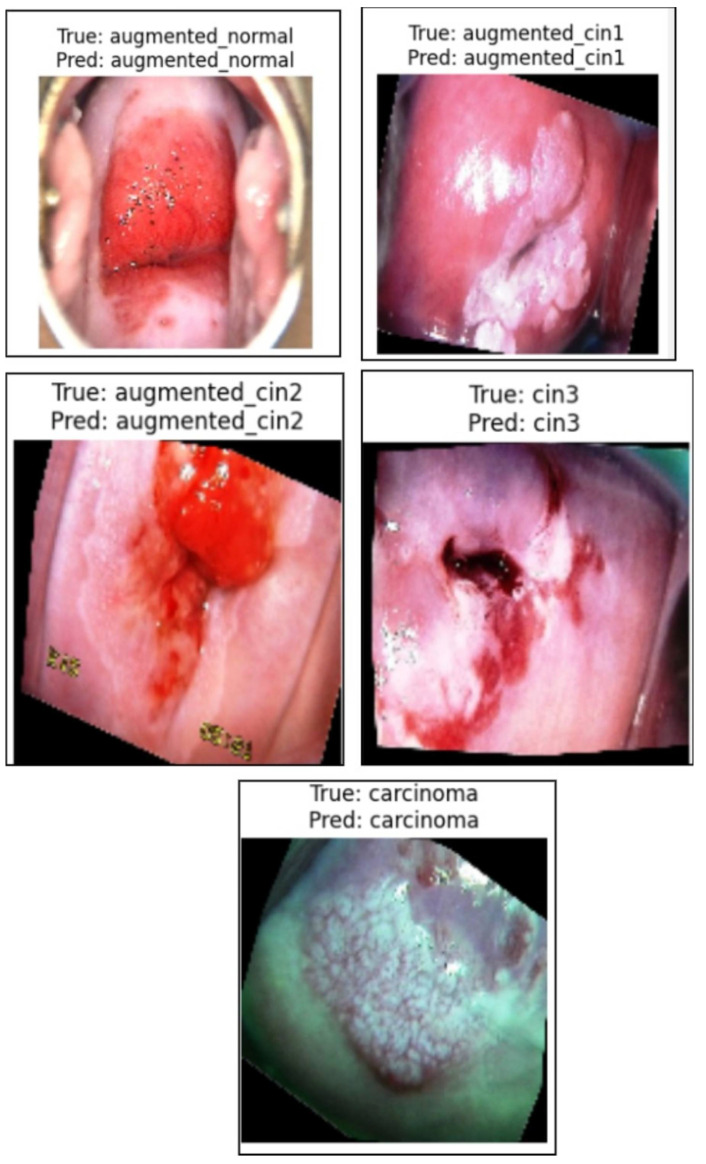
Illustration of the prediction of the five classes by the proposed algorithm from normal to carcinoma.

**Figure 8 cancers-17-01521-f008:**
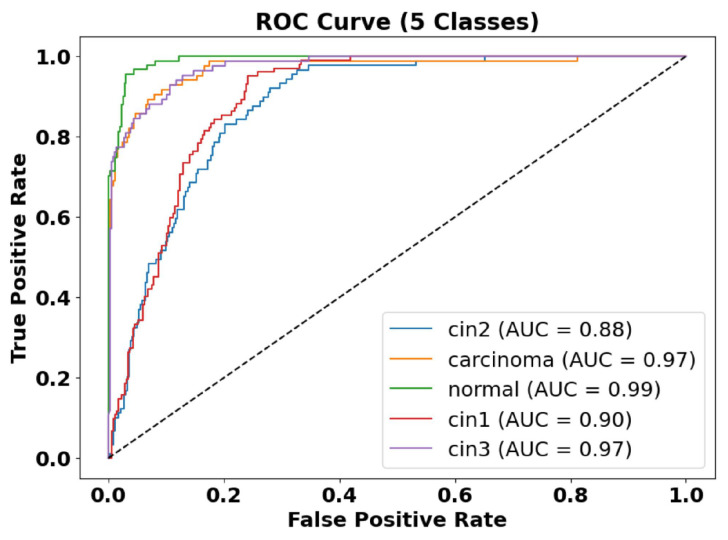
ROC-AUC curve of five classes. The AUC of the normal class is 0.99, the AUC of CIN1 is 0.90, the AUC of CIN2 is 0.88, the AUC of CIN3 is 0.97, and the AUC of carcinoma is 0.97.

**Figure 9 cancers-17-01521-f009:**
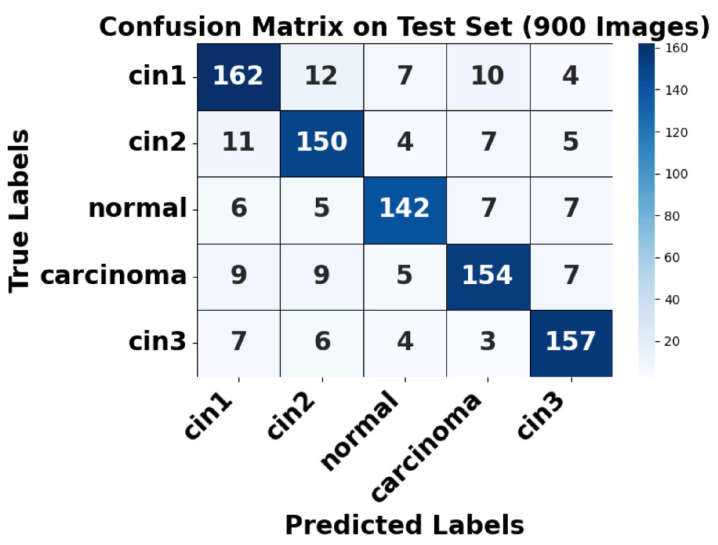
Confusion matrix of test set.

**Figure 10 cancers-17-01521-f010:**
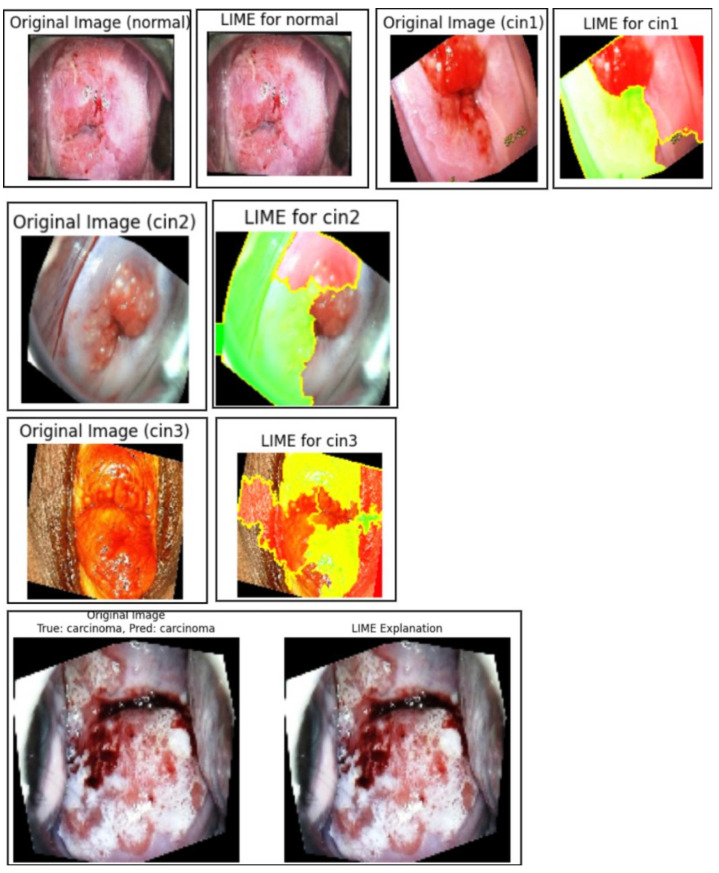
Normal, CIN1, CIN2, CIN3, and carcinoma regions of the primary dataset generated using LIME.

**Figure 11 cancers-17-01521-f011:**
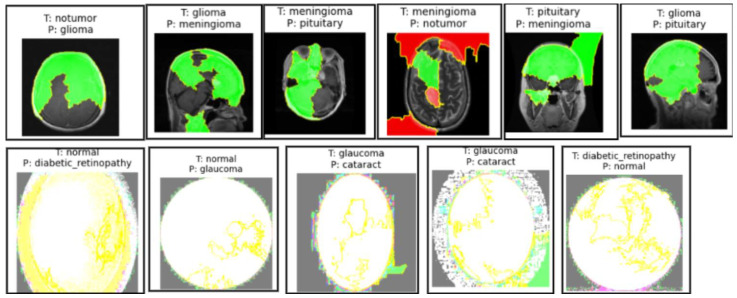
Sample images from the brain MRI dataset in the first row and the eye_disease dataset in the second row show misclassifications by the model explained by LIME.

**Figure 12 cancers-17-01521-f012:**
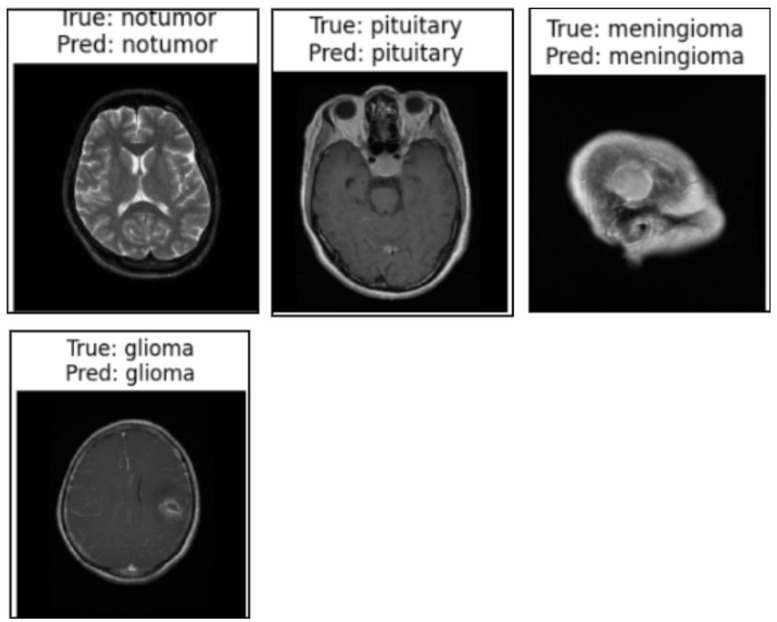
Illustration of the prediction of the four classes of the brain MRI dataset by the proposed algorithm.

**Figure 13 cancers-17-01521-f013:**
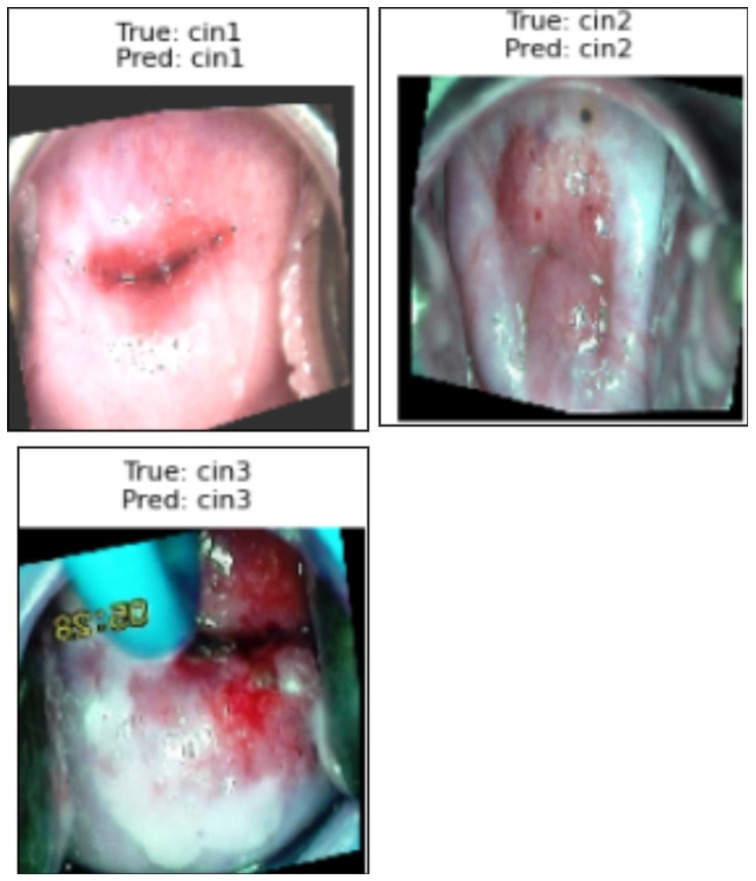
Illustration of the prediction of the three classes of the Malhari dataset by the proposed algorithm.

**Figure 14 cancers-17-01521-f014:**
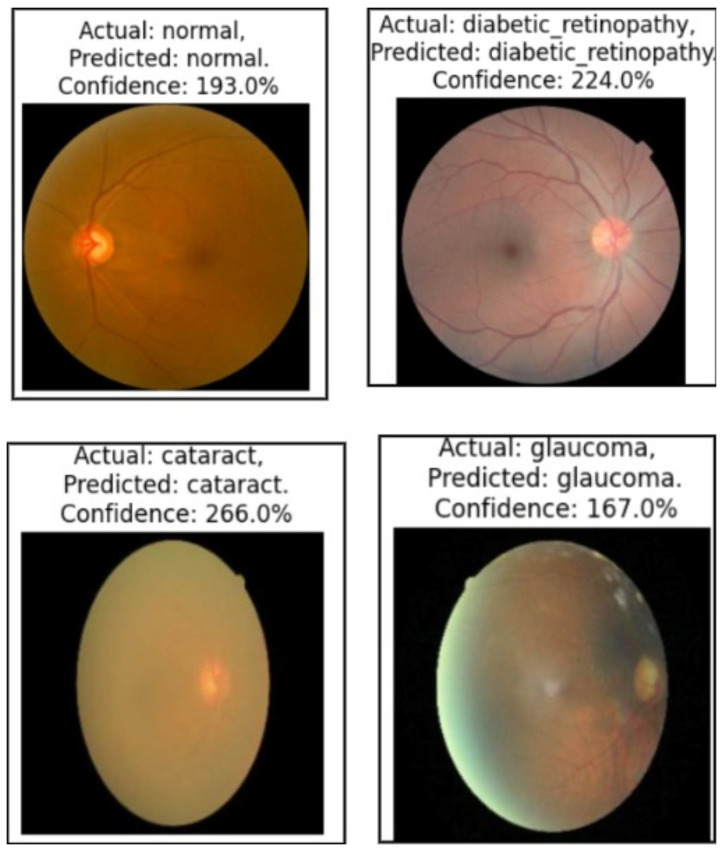
Illustration of the prediction of the four classes of the eye_disease dataset by the proposed algorithm.

**Table 1 cancers-17-01521-t001:** Parameters used in segmentation workflow.

Parameter	Value
Image Dimensions	128 × 128 pixels
Batch Size	8
Epochs	100
Learning Rate	1 × 10^−4^
Loss Function	Binary Cross-Entropy
Metrics	Dice Coefficient, IoU, Precision, Recall
Training Dataset Split	64%
Validation Dataset Split	16%
Test Dataset Split	20%

**Table 2 cancers-17-01521-t002:** Steps in graph construction and augmentation.

Stage	Description
Feature Extraction	Extracted geometric and texture features from segmented masks for five classes (normal, CIN1, CIN2, CIN3, carcinoma).
Initial Graph Construction	Created one graph per class using original features. Nodes represent segmented regions, edges are based on Euclidean distances between feature vectors, and edge weights are the inverse of distances.
Feature Augmentation	Applied Gaussian noise to features (area, perimeter, eccentricity, mean intensity, and contrast) with noise level defined as a fraction of the standard deviation.
Augmented Graph Construction	Generated 500 graphs per class using augmented features. Threshold for edge creation was randomized, and edge weights were computed using the inverse-distance formula.
Graph Storage	Saved graphs in GML format and visualized graphs using a spring layout.

**Table 3 cancers-17-01521-t003:** Overview of the sample size of the dataset used in different phases of the experiment.

Name of the Dataset	Sample Size
Primary dataset (as received from IARC)	Normal: 136, CIN1: 278, CIN2: 286, CIN3: 296, carcinoma: 200.
After augmentation (using *Albumentations* library [64])	Augmented to 900 images and 900 masks per class.
Graphs for classification (augmented dataset)	500 graphs per class considered for performing classification.
Final classification (on augmented primary dataset)	900 images, 900 masks per class, and 500 graphs per class considered.
Model generalization using secondary datasets	Malhari dataset: 900 CIN1 images, 930 CIN2 images, 960 CIN3 images.
	Brain MRI dataset: 1595 no tumor, 1457 pituitary, 1339 meningioma, 1321 glioma images.
	eye_disease dataset: 1038 cataract, 1098 diabetic retinopathy, 1007 glaucoma, and 1074 normal images.

**Table 4 cancers-17-01521-t004:** Computational efficiency: inference time, FLOPs, and GPU memory usage across different platforms.

Model	Inference Time (TPU)	Inference Time (GPU)	FLOPs (Training)	Peak GPU Memory Usage
DeepLabV3	30 ms per image	50 ms per image	∼40 GFLOPs	5.8 GB
GCN	20 ms per graph	30 ms per graph	∼2.5 GFLOPs	2.1 GB

**Table 5 cancers-17-01521-t005:** Segmentation metrics for colposcopy images across five classes.

Class	Dice Coefficient (%)	IoU (%)	Precision (%)	Recall (%)
Normal	90.5	82.1	91.0	90.0
CIN1	92.4	85.4	93.1	91.7
CIN2	91.8	84.7	92.5	91.0
CIN3	93.01	86.8	94.0	92.5
Carcinoma	92.56	86.1	93.7	91.8

**Table 6 cancers-17-01521-t006:** Geometric and texture features for different cervical cancer stages.

Feature	Class				
	Normal	CIN1	CIN2	CIN3	Carcinoma
Perimeter	150.2	120.3	130.6	140.4	155.8
Area	1800.4	900.5	1100.7	1500.9	1750.6
Circularity	0.86	0.82	0.00	0.89	0.00
Eccentricity	0.85	0.42	0.92	0.88	0.90
Contrast	310.6	400.8	450.9	520.5	600.3
Mean Intensity	0.65	0.68	0.55	0.58	0.50

**Table 7 cancers-17-01521-t007:** Comprehensive class-wise performance metrics before and after fine-tuning with ablation on noisy and low-contrast data.

Metric	Class	Before FT ($M_{100}$)	After FT ($M_{50}$)	Noisy	Low Contrast	Epochs Trained
Training Accuracy (${\mathrm{ATA}}_{100},{\mathrm{ATA}}_{50}$)	Normal	91.2	97.6	–	–	100/51
	CIN1	89.8	96.3	–	–	
	CIN2	90.5	96.8	–	–	
	CIN3	89.9	96.7	–	–	
	Carcinoma	90.1	96.9	–	–	
Training Loss (${\mathrm{ATL}}_{100},{\mathrm{ATL}}_{50}$)	Normal	0.158	0.070	–	–	100/51
	CIN1	0.170	0.073	–	–	
	CIN2	0.165	0.068	–	–	
	CIN3	0.162	0.069	–	–	
	Carcinoma	0.160	0.067	–	–	
Precision (${\mathrm{AP}}_{100},{\mathrm{AP}}_{50}$)	Normal	89.7	95.5	–	–	
	CIN1	88.5	94.1	–	–	
	CIN2	89.0	94.7	–	–	
	CIN3	89.6	95.2	–	–	
	Carcinoma	90.2	95.8	–	–	
Recall (${\mathrm{AR}}_{100},{\mathrm{AR}}_{50}$)	Normal	90.1	95.8	–	–	
	CIN1	88.7	94.3	–	–	
	CIN2	89.3	95.0	–	–	
	CIN3	89.8	95.5	–	–	
	Carcinoma	90.5	96.0	–	–	
F1-Score ($\mathrm{AF}1_{100},\mathrm{AF}1_{50}$)	Normal	89.9	95.6	91.1	89.4	100/51
	CIN1	88.6	94.2	89.5	88.1	
	CIN2	89.2	94.8	90.2	88.7	
	CIN3	89.7	95.3	90.8	89.5	
	Carcinoma	90.3	95.9	91.5	90.3	
Validation Accuracy (${\mathrm{AVA}}_{100},{\mathrm{AVA}}_{50}$)	Normal	92.3	99.1	94.2	91.6	100/51
	CIN1	91.5	98.8	92.3	90.8	
	CIN2	92.0	99.0	93.1	91.4	
	CIN3	91.8	98.9	93.7	91.9	
	Carcinoma	92.4	99.2	94.5	92.3	
Validation Loss (${\mathrm{AVL}}_{100},{\mathrm{AVL}}_{50}$)	Normal	0.152	0.050	0.125	0.149	100/51
	CIN1	0.160	0.052	0.131	0.150	
	CIN2	0.155	0.051	0.118	0.143	
	CIN3	0.158	0.050	0.120	0.140	
	Carcinoma	0.151	0.049	0.115	0.139	
Macro-Average F1-Score ($\mathrm{AF}1\_{\mathrm{Macro}}_{100},\mathrm{AF}1\_{\mathrm{Macro}}_{50}$)	All Classes	89.4	94.56	90.6	89.2	100/51
Macro-Average Validation Accuracy ($\mathrm{AVA}\_{\mathrm{Macro}}_{100},\mathrm{AVA}\_{\mathrm{Macro}}_{50}$)	All Classes	92.1	98.98	93.6	91.6	
Early Stopping Applied?	All Classes	No	Yes (No imp. for 3 epochs)	–	–	

**Table 8 cancers-17-01521-t008:** LIME evaluation metrics (class wise).

Class	Fidelity Score (%)	Stability Score (%)	Local Accuracy (%)
Normal	91.2	89.8	92.6
CIN1	89.4	87.9	91.0
CIN2	90.5	88.5	92.3
CIN3	91.8	90.1	93.7
Carcinoma	92.1	89.4	94.2
Average	91.0	89.1	92.8

**Table 9 cancers-17-01521-t009:** Segmentation and classification performance across datasets.

Dataset	Metric	Solution/Class	Label	Value
Segmentation Performance				
Malhari	Dice Coeff. (%)	Lugol’s iodine	CIN1, CIN2, CIN3	84.2, 85.8, 86.1
	Dice Coeff. (%)	Acetic acid	CIN1, CIN2, CIN3	85.9, 86.5, 86.98
	Dice Coeff. (%)	Normal saline	CIN1, CIN2, CIN3	83.5, 84.0, 85.6
	IoU (%)	Lugol’s iodine	CIN1, CIN2, CIN3	84.5, 85.1, 85.6
	IoU (%)	Acetic acid	CIN1, CIN2, CIN3	84.0, 84.6, 85.2
	IoU (%)	Normal saline	CIN1, CIN2, CIN3	83.8, 84.2, 84.7
Brain MRI	Dice Coeff (%)	–	No Tumor, Glioma, Meningioma, Pituitary	90.5, 91.8, 92.1, 92.56
	IoU (%)	–	No Tumor, Glioma, Meningioma, Pituitary	82.1, 84.7, 86.8, 86.1
Eye Disease	Dice Coeff (%)	–	Normal, DR, Cataract, Glaucoma	91.8, 92.3, 92.8, 93.56
	IoU (%)	–	Normal, DR, Cataract, Glaucoma	90.1, 90.7, 91.8, 92.1
Classification Performance (Before and After Fine-Tuning)				
Malhari	Macro F1 (%)	All	All CIN	B: 89.4, A: 90.56
	Val. Acc. (%)	All	All CIN	B: 89.88, A: 91.02
	Epochs Trained	All	All CIN	B: 100, A: 51
	Early Stopping?	All	All CIN	No → Yes
Brain MRI	Macro F1 (%)	All	All MRI	B: 88.4, A: 91.56
	Val. Acc. (%)	All	All MRI	B: 90.1, A: 91.98
	Epochs Trained	All	All MRI	B: 100, A: 51
	Early Stopping?	All	All MRI	No → Yes
Eye Disease	Macro F1 (%)	All	All Eye	B: 89.4, A: 90.56
	Val. Acc. (%)	All	All Eye	B: 92.1, A: 94.98
	Epochs Trained	All	All Eye	B: 100, A: 51
	Early Stopping?	All	All Eye	No → Yes

**Table 10 cancers-17-01521-t010:** Performance comparison of fixed-graph, focal loss, and proposed adaptive augmented graph models.

Metric	Graph with Fixed Weights	Focal Loss (No Augmentation)	Proposed Model (Augmented)
Macro-Average F1-Score (%)	86.5	88.9	94.56
Validation Accuracy (%)	88.2	91.3	98.98
Epochs Trained (Before Fine-Tuning)	100	100	100
Epochs Trained (After Fine-Tuning)	100	72	51
Early Stopping Applied?	No	Yes (Patience: 5 epochs)	Yes (No improvement for 3 epochs)

**Table 11 cancers-17-01521-t011:** Performance comparison before and after fine-tuning of the baseline models with the proposed model.

Metric	EfficientNet	ViT	ResNet50	Ours
M1	87.4/93.86	88.9/94.1	86.3/92.9	89.4/94.56
M2	90.23/97.68	90.89/98.5	90.3/96.2	92.1/98.98
Epochs Trained	100/50	100/50	100/50	100/50
Early Stopping?	No	No	No	Yes
				(No improvement for 3 epochs)

**Table 12 cancers-17-01521-t012:** Performance comparison of the proposed approach with alternative models.

Method	M1: Macro-Average F1-Score	M2: Macro-Average Validation Accuracy	Epochs Trained
Segmentation-Based Approaches			
MTANet [75]	88.9/93.7	90.5/96.2	100/50
Segment Anything Model (SAM) [78]	87.5/93.0	89.8/95.5	100/50
Feature-Extraction-Based Approaches			
Deep Learning Model for Cervical Cancer Prediction [76]	87.6/92.1	89.8/95.4	100/50
Hybrid Deep Feature Extraction [79]	88.1/93.4	90.2/96.1	100/50
Graph-Based Learning Approaches			
Graph Attention Networks (GAT) [80]	89.0/94.2	91.0/97.1	100/50
Graph Transformer Networks (GTN) [81]	89.4/94.5	91.3/97.4	100/50
K-Nearest Neighbour [8]	86.9/89.8	89.3/91.5	100/50
Hybrid Models for Classification			
CerviFusionNet [77]	89.2/94.0	91.2/96.8	100/50
Our Proposed Approach	89.4/94.56	92.1/98.98	100/51 (Early Stopping)

## Data Availability

The data presented in this study are available on request from the corresponding author. The implementation is publicly available at Graph-Based Cervical Abnormality Classification.

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
