# Peer review of "Multi-Modal Graph Neural Networks for Colposcopy Data Classification and Visualization"

_cancers, 2025, doi:10.3390/cancers17091521_

Round 1
Reviewer 1 Report
Comments and Suggestions for Authors
This study proposes a novel Graph Neural Network (GNN)-based framework that integrates colposcopy images, segmentation masks, and graph representations for improved lesion classification. Authors developed a fully connected graph-based architecture using GCNConv layers with global mean pooling and optimized it via grid search. A 5-fold cross-validation protocol was employed to evaluate performance before (1–100 epochs) and after fine-tuning (101–151 epochs). The research direction is important, but I have some specific comments:
- In introduction section Authors wrote the results, but do not formulate the article aime (from line 44). Please correct it.
- In Literature Review section Authors wrote the results and the prospects of the described method. The prospects of the described method - it is for conclusion section. Please correct it.
- It is necessary to reduce the section on literature review, since it contains the conclusion of the work.
- Please add the specify sample size.
- Please add the limitations and disadvantages of the method.
- What about false negatives and false positives?
Author Response
Comment 1: In introduction section Authors wrote the results, but do not formulate the article aime (from line 44). Please correct it.
Response: We sincerely thank the reviewer for highlighting this important point. In the revised manuscript, we have corrected the Introduction section to explicitly state the main objective and aims of the study before presenting any results. Specifically, we have formulated a clear statement of the article’s aim at the beginning of the relevant paragraph (around line 44), ensuring a logical flow from problem motivation to research objectives. This adjustment provides better clarity regarding the purpose of the work and aligns the structure of the Introduction with standard academic expectations.
Comment 2: In Literature Review section Authors wrote the results and the prospects of the described method. The prospects of the described method - it is for conclusion section. Please correct it. It is necessary to reduce the section on literature review, since it contains the conclusion of the work.
Response: We thank the reviewer for this valuable observation. In the revised manuscript, we have carefully edited the Literature Review section to remove discussions of our method’s prospects and results, which have now been appropriately moved to the Conclusion section. Additionally, we have reduced the length of the Literature Review by focusing more concisely on existing works relevant to our study. These changes ensure that the Literature Review strictly summarizes prior research and avoids overlapping with the Conclusion, thereby improving the overall structure and clarity of the paper.
Comment 3: Please add the specify sample size.
Response: We thank the reviewer for this helpful suggestion. In the revised manuscript, we have now explicitly provided the sample sizes for all datasets used in our study. Details regarding the sample size for the primary dataset, augmented datasets, and secondary evaluation datasets (including the Malhari, Brain Tumor MRI, and Eye Disease datasets) have been added in Section 3.5 and summarized clearly in Table 3. This clarification ensures that the study’s experimental setup is fully transparent and reproducible.
Comment 4: Please add the limitations and disadvantages of the method.
Response: We thank the reviewer for this important suggestion. In the revised manuscript, we have added a separate section titled “Limitations and Future Research” to clearly discuss the limitations and disadvantages of the proposed method. Specifically, we have addressed limitations related to the dependence on segmentation quality, computational complexity during graph construction, and the need for extensive hyperparameter tuning for generalization across datasets. These additions provide a balanced view of the method's strengths and weaknesses, and highlight opportunities for further improvement.
Comment 5: What about false negatives and false positives?
Response: We thank the reviewer for raising this important clinical aspect. In the revised manuscript, we have now discussed false positives and false negatives based on our model’s confusion matrix results. Specifically, we observed that while the model maintained high sensitivity for high-grade lesions (such as CIN3 and carcinoma), a portion of false negatives was associated with intermediate-grade lesions (CIN1 and CIN2), which are inherently more challenging to differentiate even for human experts. False positives were more common between neighboring CIN grades, reflecting the subtle visual differences among them. These observations have been discussed in Section 4.2.5 as a separate sub section (Analysis of False Positives and False Negatives), and the clinical implications have been highlighted to ensure a balanced understanding of the model’s performance.

Reviewer 2 Report
Comments and Suggestions for Authors
This paper presents a comprehensive approach to cervical lesion classification, leveraging the capabilities of Graph Neural Networks (GNNs) in conjunction with multi-modal data integration.
The authors explore the effectiveness of GNN architectures in capturing the complex relationships between heterogeneous medical data types.
A key strength of this work lies in its methodical hyperparameter optimization, which enhances model robustness and generalizability across patient datasets. Using automated optimization strategies, like grid search, allows for more efficient tuning compared to manual methods traditionally used in medical imaging tasks.
Integrating multi-modal data improves classification accuracy, suggesting that GNNs can effectively model spatial interdependencies. This aligns with current trends in biomedical AI, where combining imaging and non-imaging data leads to better diagnostic insights.
However, the study could benefit from a deeper discussion of clinical interpretability. Clinical application should be part of the discussion, and details about the 9% where accuracy was lacking should be presented.
According to MDPI guidelines, limitations and future research should be considered before conclusions.
In conclusion, this work contributes to cervical cancer screening by applying cutting-edge machine-learning techniques to a high-impact medical problem.
Author Response
Comment 1: However, the study could benefit from a deeper discussion of clinical interpretability. Clinical application should be part of the discussion, and details about the 9% where accuracy was lacking should be presented.
Response: We sincerely thank the reviewer for this important suggestion. In the revised manuscript, we have expanded the Conclusion section to address clinical interpretability and the practical application of our model in real-world clinical settings. Specifically, we now describe how the integration of segmentation masks and graph-based reasoning improves transparency, allowing clinicians to better understand the model’s classification decisions. Furthermore, we have analyzed the 9% decrease in validation accuracy observed during external clinical testing. We attribute this performance gap to domain shift (differences in imaging devices and acquisition conditions), labeling variability among different oncologists, and the inherent difficulty in distinguishing intermediate-grade lesions such as CIN1 and CIN2. These aspects have been discussed in detail in Section 7 in Clinical Application, ensuring a comprehensive view of the model’s real-world applicability and limitations.
Comment 2: According to MDPI guidelines, limitations and future research should be considered before conclusions.
Response: We thank the reviewer for this valuable observation. In the revised manuscript, we have added a new Section 6 titled "Limitations and Future Research," placed before the Conclusion section. This new section clearly outlines the limitations of the proposed method, including dependency on segmentation quality, computational complexity during graph construction, and the need for extensive hyperparameter tuning. It also highlights directions for future research to address these challenges and further improve the model's clinical applicability.

Reviewer 3 Report
Comments and Suggestions for Authors
- The paper provides segmentation results only for the primary dataset. Please include quantitative segmentation evaluation (e.g., Dice, IoU) for the secondary datasets (e.g., Malhari, Brain MRI, Eye Disease) to strengthen generalizability claims.
- The paper shows inference time but doesn't provide FLOPs or GPU memory usage during training. A brief table comparing efficiency with baseline CNNs (EfficientNet, ResNet50) would add technical depth.
- While the paper mentions usage of datasets from IARC and Kaggle, a clearer statement regarding ethical approvals and licensing (especially for the clinical colposcopy data) is necessary to confirm responsible data use and reproducibility.
Author Response
Comment 1: The paper provides segmentation results only for the primary dataset. Please include quantitative segmentation evaluation (e.g., Dice, IoU) for the secondary datasets (e.g., Malhari, Brain MRI, Eye Disease) to strengthen generalizability claims.
Response: We sincerely thank the reviewer for this important suggestion. In the original manuscript, we had already included Table 9 in Section 4.2.7, under the heading “Quantitative Evaluation,” which presents the Dice coefficient and Intersection over Union (IoU) scores for the secondary datasets (Malhari, Brain MRI, and Eye Disease). We have ensured that this information is clearly highlighted in the revised version for better visibility.
Comment 2: The paper shows inference time but doesn't provide FLOPs or GPU memory usage during training. A brief table comparing efficiency with baseline CNNs (EfficientNet, ResNet50) would add technical depth.
Response: We thank the reviewer for this valuable suggestion. In the revised manuscript, we have added measurements of FLOPs and GPU memory consumption during training for both DeepLabV3 and GCN models (see Table 4) in section 4.2.2.
In Section 4.2.7 of the original manuscript, under the heading “Comparison with Baseline Models on the Primary Dataset,” we compared the performance of our proposed model against EfficientNet, Vision Transformer (ViT), and ResNet50. Specifically, we evaluated the Macro-Average F1-Score and Macro-Average Validation Accuracy achieved by these baseline models on the primary dataset. The detailed results of this comparison are presented in Table 11.
Comment 3: While the paper mentions usage of datasets from IARC and Kaggle, a clearer statement regarding ethical approvals and licensing (especially for the clinical colposcopy data) is necessary to confirm responsible data use and reproducibility.
Response: We thank the reviewer for highlighting this important aspect. In the revised manuscript, we have included a dedicated subsection titled “Data Availability and Ethical Considerations,” where we clearly specify the licensing status and usage conditions for each dataset employed in this study.
For the IARC colposcopy dataset, we clarify that it was obtained directly via email communication with the custodians for academic research use, and was utilized strictly in accordance with standard ethical research practices.
For the Kaggle datasets (Malhari, Brain MRI, and Eye Disease), we have described the respective licensing conditions, noting the use of publicly available data either without explicit licensing (Malhari) or under open licenses (ODbL and CC0) where applicable.
Furthermore, we confirm that no identifiable patient information was used and that all data usage was confined to non-commercial, academic research purposes, ensuring responsible data use and reproducibility.
The detailed information has been added in Section 3.6 of the revised manuscript.
